# DecAEvolve: Decompose, Adapt, and Evolve, or, Three Pillars of Effective LLM-based Scientific Equation Discovery

## Abstract

Finding mathematical relations underlying natural phenomena and scientific systems has been one of the fundamental tasks in the history of scientific discovery. Recent advancements in evolutionary search with Large Language Models (LLMs), with their embedded scientific knowledge, have shown great promise for this task. However, discovering such mathematical models governing scientific observations still remains significantly challenging, as it requires navigating vast combinatorial hypothesis spaces with an explosion of possible relations. Existing LLM-based approaches overlook the impact of data on the structure of mathematical relations, and treat LLMs as a static hypothesis generator unaware of the observed scientific system. This leads to suboptimal and inefficient exploration of the hypothesis space with over-reliance on LLMs' internal priors. To bridge this gap, we introduce *Decompose, Adapt, and Evolve* (**DecAEvolve**), a framework that leverages granular feedback from symbolic term decomposition and LLM refinement through reinforcement learning (RL) fine-tuning to enhance both robustness and efficiency of evolutionary discovery frameworks. DecAEvolve unifies symbolic decomposition with test-time RL adaptation, enabling adaptive rather than static hypothesis generation and reducing error by up to an order of magnitude compared to state-of-the-art baselines. Our experiments across diverse datasets demonstrate that DecAEvolve significantly improves the accuracy of discovered equations and the efficiency of the discovery process compared to the state-of-the-art baselines[1].

## 1 Introduction

The emergence of Large Language Models (LLMs) has fundamentally transformed automated problem-solving across diverse domains. Beyond their well-established capabilities in natural language understanding and programming (Achiam et al., 2023; Touvron et al., 2023), LLMs have recently demonstrated remarkable reasoning abilities that enable them to tackle complex optimization and discovery tasks. Their capacity to leverage embedded domain knowledge, interpolate between them, generate structured hypotheses and engage in iterative refinement, positions LLMs as powerful engines for systematic exploration of complex solution spaces towards discovery goals (Romera-Paredes et al., 2024; Novikov et al., 2025; Surina et al., 2025). This potential extends naturally to scientific discovery tasks, where the combination of domain expertise and systematic search/exploration in the hypothesis space can unlock new approaches to longstanding challenges of scientific inquiry (Shojaee et al., 2025a).

Scientific equation discovery—the process of uncovering compact and interpretable mathematical models that govern natural phenomena—represents one of the fundamental tasks in automated scientific discovery, with applications across many fields of science such as physics, biology, and material science (Makke & Chawla, 2024). Traditional approaches in Symbolic Regression (SR) rely on genetic programming and evolutionary strategies (Koza, 1994b; Cava et al., 2021b); however, these approaches often struggle with scalability limitations and inefficient exploration of the vast combinatorial hypothesis space (Virgolin & Pissis, 2022). More recent advances have introduced neural-guided approaches, where deep learning architectures are trained to generate or refine symbolic expressions

---

[1]Code is available at: `https://anonymous.4open.science/r/decaevolve-1215`

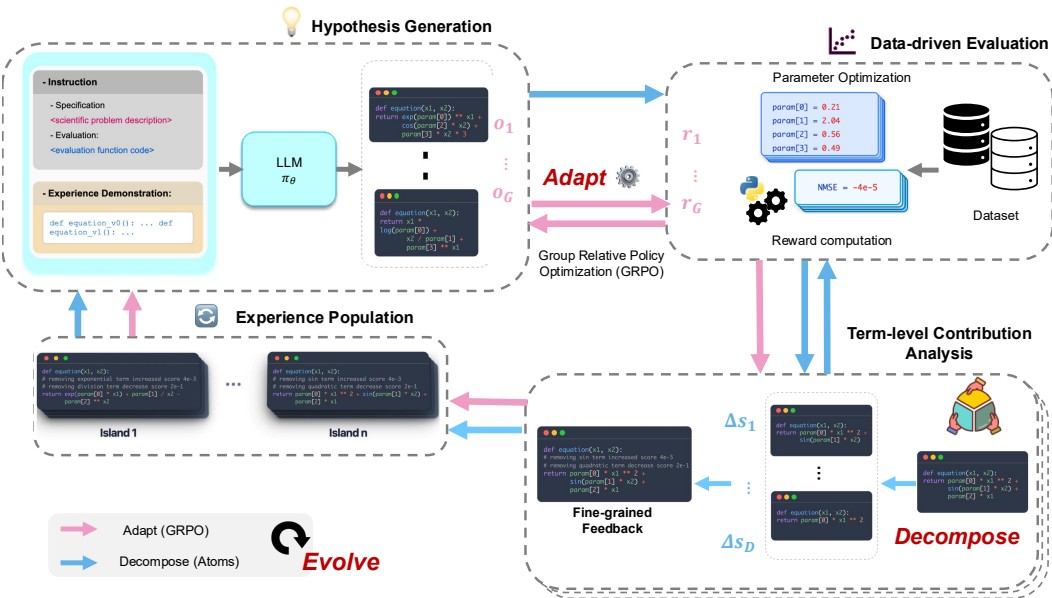

Figure 1: **Overview of the DecAEvolve framework.** The framework integrates *Adaptation* (LLM fine-tuning via reinforcement learning using Group Relative Policy Optimization with data-driven rewards) and *Decomposition* (granular-level feedback through symbolic atomic term analysis) within an *Evolutionary* search process. The adaptation aligns the LLM to the target scientific system beyond its internal priors, while decomposition provides fine-grained guidance for hypothesis refinement. Iterating these three key components enables effective and efficient exploration of the combinatorial hypothesis space in equation discovery.

(Udrescu & Tegmark, 2020b; Bruneton, 2025a), and transformer-based methods that are pre-trained with large-scale synthetic data to directly model symbolic sequences as language generation tasks (Kamienny et al., 2022a; Shojaee et al., 2023a; Meidani et al., 2024b). These developments have demonstrated promising capabilities in data-driven learning, yet are limited in balancing learning and search components and in incorporating scientific prior knowledge into the process of discovery.

Several works have recently introduced promising frameworks to integrate LLMs for scientific equation discovery, leveraging their scientific priors and reasoning capabilities to navigate the complex landscape of mathematical expressions more effectively. Notably, LLM-SR (Shojaee et al., 2025a) combines LLMs' scientific knowledge with multi-island evolutionary search, generating equation hypotheses as Python function skeletons guided by data feedback. LaSR (Grayeli et al., 2024b) introduces a concept learning approach that extracts abstract textual concepts from successful equation hypotheses, using these concepts to guide both evolutionary search (with PySR (Cranmer, 2023)) and LLM-based hypothesis generation. SGA (Ma et al., 2024b) employs a bilevel optimization framework that iteratively combines LLMs for discrete hypothesis generation with physical simulations for continuous parameter optimization. These methods demonstrate this potential by combining LLMs' domain expertise with systematic search strategies, treating equation discovery as a program synthesis problem guided by scientific knowledge (Shojaee et al., 2025b; Reddy & Shojaee, 2025).

Our key insight is that equation discovery benefits from both adaptation (aligning the model with data distributions) and decomposition (understanding which symbolic components matter), neither of which prior LLM frameworks integrate. While classical methods such as SINDy (Brunton et al., 2016) employ sparse regression techniques that implicitly decompose equations through term selection, they lack the adaptive learning and scientific reasoning capabilities that LLMs provide. Current LLM-based discovery methods also exhibit fundamental limitations that constrain their effectiveness. First, they treat LLMs as static hypothesis generators, where the model's parameters remain fixed regardless of the problem domain, nuances of the specific observed system or, insights gained during the search process. This prevents LLMs from adapting their generation strategies based on the specific problem, the data, and the domain-specific requirements. Second, existing approaches mainly provide coarse-grained feedback about solution quality, typically limited to scalar reward signals (e.g., mean squared error) from execution of whole hypothesis that indicate which hypotheses perform well respectively, without revealing why specific mathematical components or patterns drive success. This

limited feedback mechanism prevents LLMs from understanding the underlying symbolic structure of successful solutions and refining their search strategies accordingly.

To address these limitations, we introduce **DecAEvolve** (Decompose, Adapt, and Evolve), a novel framework that enhances the effectiveness and efficiency of LLM-based equation discovery by combining decomposition-based feedback with test-time adaptation within an LLM-guided evolutionary framework. Our key contributions are as follows:

- We develop a systematic methodology for providing LLMs with interpretable directional feedback about which components of their generated hypothesis prove effective. Through structured hypothesis decomposition and evaluations, the contributions of individual terms and their interactions are quantified and provided as feedback. This enables LLMs to understand not just which hypotheses succeed, but *why* specific mathematical building blocks are effective, transforming blind generation into informed iterative refinement.

- We employ reinforcement learning with Group-Relative Policy Optimization (GRPO) to implicitly distill the data distribution into the model's parameters for better hypothesis generation process. This test-time adaptation/training approach allows the LLM to learn from successful equation discoveries without directly observing raw data, progressively aligning its hypothesis generation with the underlying symbolic relationships through reward-weighted gradient updates.

- We demonstrate that these synergistic contributions dramatically improve search efficiency, requiring significantly fewer iterations to discover accurate symbolic expressions. Our comprehensive evaluation across multiple benchmarks shows superior performance compared to LLM-SR and other baselines in both in-domain and out-of-domain settings, validating the effectiveness of our guided discovery approach.

## 2 PRELIMINARIES

**Problem Formulation.** In scientific equation discovery, the goal is to find a compact mathematical expression (hypothesis) $h(\mathbf{x}; \mathcal{T})$ that approximates an unknown target function $f : \mathbb{R}^d \to \mathbb{R}$ within the context of a specific scientific problem $\mathcal{T}$ with dataset of input–output pairs $\mathcal{D} = \{(\mathbf{x}_i, y_i)\}_{i=1}^n$. The objective is to discover functional relationships such that $f(\mathbf{x}_i) \approx y_i$ for all $i$, producing expressions that are both interpretable and capable of generalizing to unseen data. Performance is typically evaluated using fitness to data with metrics such as mean squared error: $\text{MSE}(h, \mathcal{D}) = -\frac{1}{n} \sum_{i=1}^n \left( h(\mathbf{x}_i; \mathcal{T}) - y_i \right)^2$.

**LLM-SR Framework.** A recent advance in this space is LLM-SR (Shojaee et al., 2025a), which pioneers reframing of symbolic regression as a program synthesis task. Instead of traditional expression trees, equations are represented as executable Python functions with placeholder parameters. The framework leverages large language models (LLMs) to iteratively generate equation program skeletons, drawing on their embedded scientific priors (i.e., built-in knowledge about a scientific problem without explicit task-specific feedback or fine-tuning) as well as programming, and reasoning capabilities. In LLM-SR, each proposed equation skeleton first undergoes data-driven parameter optimization (via BFGS or Adam), and then, high-scoring hypotheses are stored in a dynamic multi-island experience buffer to guide the program optimization. Subsequent LLM prompts incorporate these top hypotheses as in-context examples sampled from buffer, enabling iterative refinement of the search process. More details on LLM-SR, its evolution, multi-island buffer design, and sampling strategies are provided in Appendix A. Our framework DecAEvolve builds on LLM-SR foundations and integrate its evolutionary search mechanism with decomposition-based feedback and test-time training. This guides the LLM beyond its default priors toward regions of hypothesis space that better align with the actual observed scientific system, enabling more effective and efficient discovery.

**Group-Relative Policy Optimization (GRPO).** GRPO (Shao et al., 2024) is an effective reinforcement learning technique that has recently gained attention to fine-tune LLMs with verifiable outcome rewards towards enhancing models' reasoning capabilities. The key innovation of GRPO is its use of group-relative advantages computed from multiple completions per prompt to provide stable reward signals. For each prompt $p$ with completions $\{h_i\}_{i=1}^G$ from old policy $\pi_{\theta_{old}}$ and corresponding rewards $\{r_i\}_{i=1}^G$, the method calculates group-relative advantages $A_i = r_i - b(p)$ where $b(p) = \frac{1}{G} \sum_{j=1}^G r_j$ serves as the per-prompt baseline. This per-prompt normalization across group samples provides variance reduction and stable credit assignment across candidates. The GRPO objective optimizes:

$$\mathcal{L}(\theta) = - \, \mathbb{E}_{p, \{h_i\} \sim \pi_{\theta_{\text{old}}}}$$

$$\frac{1}{G} \sum_{i=1}^{G} \frac{1}{|h_i|} \sum_{t=1}^{|h_i|} \Big\{ \min \Big[ \frac{\pi_\theta(h_{i,t}|p, h_{i,<t})}{\pi_{\theta_{\text{old}}}(h_{i,t}|p, h_{i,<t})} A_{i,t}, \text{clip}\Big( \frac{\pi_\theta(h_{i,t}|p, h_{i,<t})}{\pi_{\theta_{\text{old}}}(h_{i,t}|p, h_{i,<t})}, 1-\epsilon, 1+\epsilon \Big) A_{i,t} \Big]$$

$$+ \beta \, \mathbb{E}_p \big[ KL(\pi_\theta(\cdot|p) \, \| \, \pi_{\text{ref}}(\cdot|p)) \big] \Big\}, \tag{1}$$

where $\pi_{\text{ref}}$ is the frozen reference model, $\beta > 0$ controls the KL regularization strength, and $\epsilon$ controls the clipping ratio to avoid excessive single-step updates to the policy. This approach enables efficient adaptation to task-specific rewards while maintaining model stability.

## 3 METHOD

**DecAEvolve** (Decompose, Adapt, and Evolve) combines adaptation, decomposition, and evolutionary search. Adaptation employs GRPO to align the LLM with data-driven rewards, while decomposition delivers term-level feedback that highlights which symbolic components drive accuracy. Coupled with evolutionary search, these mechanisms form a feedback-driven loop that improves both equation quality and the model's generation policy. We illustrate the pseudocode for **DecAEvolve** in Algorithm 1 and emphasize its two key contributions: (i) reinforcement-learning-based test-time adaptation and (ii) fine-grained decomposition feedback.

### 3.1 TEST-TIME ADAPTATION WITH GRPO

In the adaptation stage of DecAEvolve, the generation policy $\pi_\theta^n$ is updated to improve its ability to propose valid and accurate symbolic equations which are better aligned with the observed data. After each iteration of generating hypothesis group samples and computing their corresponding rewards, the policy is fine-tuned using GRPO (Shao et al., 2024) objective function (equation 1) on the accumulated dataset $(\mathbf{p_n}, \{h_i^n, r_i^n\}_{i=1}^G)$, where each prompt at online iteration $n$ ($\mathbf{p_n}$) is paired with multiple corresponding hypothesis completions $\{h_i^n\}$ and their data-driven rewards $\{r_i^n\}$ as: $\{r_i^n\}_{i=1}^G \leftarrow \text{Score}_\mathcal{T}(\{h_i^n\}_{i=1}^G, \mathcal{D})$. The score function $\text{Score}_\mathcal{T}(\cdot)$ for each equation candidate is the negative mean squared error (MSE) with respect to the training data, which is also transformed to a bounded reward between 0 and 1 via exponential transformation: $\text{Score}_\mathcal{T}(h, \mathcal{D}) = \exp(-\text{MSE}(h, \mathcal{D}))$. Failed or invalid completions receive a floor reward of 0.01. Fine-tuning is performed using LoRA adapters, enabling efficient parameter updates while maintaining the base model as a reference anchor ($\pi_{\text{ref}}$). The KL coefficient $\beta$ in equation 1 ensures the fine-tuned model retains its general capabilities while effectively adapting to the observed scientific system with the help of data-driven reward through GRPO. More implementation details and hyperparameters are provided in Section 4 andAppendix C.

Notably, the use of GRPO in DecAEvolve differs fundamentally from its use in prior RL-tuning for reasoning literature. Here, GRPO is not used for global model fine-tuning, but for per-system, test-time adaptation that steers the model parameters toward hypotheses better aligned with the observed data of a scientific system. The equation program synthesis optimization is implicitly guided with the reward from how well an equation hypothesis candidate explains the observed data, not from correctness or textual feedback as in typical GRPO setups. Also, the inputs/prompts used in test-time adaptation GRPO here correspond to a fixed scientific system but also include dynamic in-context examples (with decomposition signal) that are sampled from the buffer in an online manner over GRPO iterations. This design helps to align model adaptation naturally with the decomposition-guided evolutionary search happening later during inference.

### 3.2 DIRECTIONAL FEEDBACK WITH TERM-LEVEL CONTRIBUTION

At the heart of our framework is an iterative discovery process in which the LLM performs a form of self-reflection: it not only generates candidate symbolic equations but also receives feedback about what certain symbolic components succeed or fail. Rather than relying solely on coarse error scores, we introduce a decomposition-based contribution analysis within the self-reflection procedure that quantifies the role of each term and its pairwise interactions, producing interpretable signals that guide subsequent iterations of discovery. A related use of decomposition has also been explored in Liu et al. (2025) for experimental-chemistry hypothesis discovery. Check Appendix D for detailed discussion.

During the decomposition step, each candidate equation program is parsed into an abstract syntax tree (AST) and decomposed into atomic symbolic terms $\{u_m(x)\}_{m=1}^M$ where $M$ is the total number of terms (see Appendix. B.1 for details). To assess the contribution of a given term $u_i$ (or a pair

---

**Algorithm 1.** DecAEvolve

---

**Input:** LLM $\pi_\theta$; dataset $\mathcal{D}$; task $\mathcal{T}$; Number of GRPO steps $N$; Group size $G$; Evolution iterations $T$; in-context size $k$; samples per prompt $b$

**Output:** Best equation $h^*$ with score $s^*$

```
# Stage 1:  Test-time adaptation with GRPO and decomposition
```
Initialize buffer: $\mathcal{B} \leftarrow \emptyset$
$h^* \leftarrow$ null, $s^* \leftarrow -\infty$
**for** $n = 1$ **to** $N$ **do**
    $\mathcal{H}_n \leftarrow \{h_j\}_{j=1}^b$ where $h_j \sim \pi_\theta(\cdot \mid \mathcal{T})$
    **for** $h \in \mathcal{H}_n$ **do**
        $s \leftarrow \text{Score}_{\mathcal{T}}(h, \mathcal{D})$
        **if** $s > s^*$ **then**
           $h^* \leftarrow h, s^* \leftarrow s$
        $\{u_m\} \leftarrow \text{Decompose}(h)$
        **for** *each term* $u_m$ **do**
           $c_m \leftarrow \text{TermContribution}(u_m, h, s, \mathcal{D})$
        $h_{\text{ann}} \leftarrow \text{Annotate}(h, \{u_m, c_m\})$
        $\mathcal{B} \leftarrow \mathcal{B} \cup \{(h_{\text{ann}}, s)\}$
    $\mathbf{p_n} \leftarrow \text{MakeFewShotPrompt}(\mathcal{B}, k)$
    $\{h_i^n\}_{i=1}^G \sim \pi_\theta(\cdot \mid \mathbf{p_n})$
    $\{r_i^n\}_{i=1}^G \leftarrow \text{Score}_{\mathcal{T}}(\{h_i^n\}_{i=1}^G, \mathcal{D})$
    $\pi_\theta \leftarrow \text{GRPO\_Update}(\pi_\theta, \{r_i^n\}_{i=1}^G)$

```
# Stage 2:  Evolutionary search with decomposition
```
$\mathcal{P}_0 \leftarrow \mathcal{B}$
**for** $t = 1$ **to** $T$ **do**
    $\mathbf{p_t} \leftarrow \text{MakeFewShotPrompt}(\mathcal{P}_{t-1}, k)$
    $\mathcal{H}_t \leftarrow \{h_j\}_{j=1}^b$ where $h_j \sim \pi_\theta(\cdot \mid \mathbf{p_t})$
    **for** $h \in \mathcal{H}_t$ **do**
        $s \leftarrow \text{Score}_{\mathcal{T}}(h, \mathcal{D})$
        **if** $s > s^*$ **then**
           $h^* \leftarrow h, s^* \leftarrow s$
        $\{u_m\} \leftarrow \text{Decompose}(h)$
        **for** *each term* $u_m$ **do**
           $c_m \leftarrow \text{TermContribution}(u_m, h, s, \mathcal{D})$
        $h_{\text{ann}} \leftarrow \text{Annotate}(h, \{u_m, c_m\})$
        $\mathcal{P}_t \leftarrow \mathcal{P}_{t-1} \cup \{(h_{\text{ann}}, s)\}$

**Output:** $h^*$ and $s^*$

---

$(u_i, u_j)$), we construct ablated hypotheses, denoted $f \backslash u_i(x)$ or $f_{\backslash u_i, u_j}(x)$, by removing the corresponding subtree from the AST and generating a new equation program in which that component no longer participates in the computation. After this symbolic ablation, we re-optimize all remaining parameters of the ablated equation on the training dataset $D$. This comparison, with each structure re-optimized to its own best-fit parameters, ensures that the performance change is attributable to the structure differences rather than suboptimal tuning and parameterization. We then quantify the contribution of a term by $\Delta_{u_i} = \text{Score}_{\mathcal{T}}(f, D) - \text{Score}_{\mathcal{T}}(f_{\backslash u_i}, D)$ and similarly for pairs $\Delta_{u_i, u_j} = \text{Score}_{\mathcal{T}}(f, D) - \text{Score}_{\mathcal{T}}(f_{\backslash \{u_i, u_j\}}, D)$). These computed contribution signals are then serialized directly into their corresponding programs as *inline comments* (without affecting executability) and stored in the buffer for evolving populations. In the next iteration, the LLM samples from this population, and the decomposition-annotated programs are reused as in-context examples. This design ensures that the model is not only guided by global error metrics but also reflects on explicit evidence of which symbolic building blocks mattered, progressively refining its generation strategy. For more details about the directional feedback mechanism and its implementation, check Appendix. B.

Algorithm 1 presents the summarized pseudo-code of DecAEvolve. The framework integrates decomposition throughout the discovery process, beginning with Stage 1 where test-time adaptation combines GRPO fine-tuning with decomposition feedback: the LLM generates candidate hypotheses, which are decomposed into symbolic terms, annotated with term-level contributions, and stored in buffer $\mathcal{B}$ to create decomposition-augmented examples for subsequent GRPO updates. The Decompose(.) and TermContribution($u_m, \ldots$) functions refer to term decomposition and contribution score estimation defined in Section 3.2; and the Annotate(.) and MakeFewShotPrompt(.) functions refer to the prompt updates with decomposition annotations (as in Figure 4), and in-context examples sampled from buffer (as in (Shojaee et al., 2025a)), respectively. Following adaptation, Stage 2 performs evolutionary search with decomposition by initializing population $\mathcal{P}_0$ from the buffer and iteratively: (i) constructs prompts with in-context few-shot examples from $\mathcal{P}_{t-1}$, (ii) generates $b$ candidate hypotheses from the adapted LLM, and (iii) evaluates, decomposes, annotates with term-level feedback, and updates the population. This unified approach leverages decomposition for both adaptation and evolutionary refinement, enabling efficient exploration of the equation space through the synergy of decomposition-guided adaptation and evolutionary search.

## 4 EXPERIMENTS

We evaluate DecAEvolve on benchmark datasets for LLM-based scientific equation discovery from (Shojaee et al., 2025a), covering domains like physics, biology, and materials science:

**Nonlinear Oscillator:** Simulates two nonlinear damped oscillators (*Oscillator1*, *Oscillator2*) governed by second-order differential equations in displacement and velocity. Both systems are designed with complex but solvable nonlinear structures that differ from standard oscillator models to challenge LLMs towards discovery through data-driven reasoning.

**Bacterial Growth:** Models E. coli growth under varying conditions of density, substrate, temperature, and pH. Novel nonlinear terms designed for temperature and pH introduce complexities that require exploration and discovery and are hard to recover from LLM recall.

**Stress-Strain Behavior:** Captures tensile response of aluminum alloy across temperatures. This dataset uses experimental measurements, providing a more realistic setting with experimental data that challenge LLM-based models beyond synthetic formulations.

All datasets consist of predefined train, validation, in-domain (ID) test, and out-of-domain (OOD) test splits. In our experiments, the training split is used for parameter optimization for each equation skeleton, the validation split is used to compute the feedback score that guides the search in symbolic space, and the held-out ID and OOD test splits are used solely for evaluation. We compare DecAEvolve against state-of-the-art non-LLM symbolic regression (SR) baselines, including approaches such as GPlearn[2], PySR[3] (Cranmer, 2023), and SINDy (Brunton et al., 2016), deep learning methods like DSR (Petersen et al., 2021) and uDSR (Landajuela et al., 2022), and pre-trained Transformer SR models NeSymReS (Biggio et al., 2021) and E2E (Kamienny et al., 2022b) (check Appendix E for implementation details). In addition, we evaluate against the leading LLM-based SR baseline, LLM-SR (Shojaee et al., 2025a), under same configurations: 3,000 LLM calls per problem with sampling temperature $\tau = 0.8$. In both approaches, equation parameters are optimized with the BFGS solver from SciPy python library and a 30s timeout used for the execution of each hypothesis. In the GRPO adaptation phase, we use batch size of 16 per device, gradient accumulation 4, learning rate $10^{-5}$, and KL coefficient $\beta = 0.05$. For fine-tuning, we use LoRA adapters with $r = 16$. Decomposition analysis is also conducted based on the AST extracted from the equation program to define each term and pairwise term contributions. We conduct experiments on six open-source models (Qwen2.5-1.5B and Qwen2.5-3B, Qwen2.5-7B, Llama-3.2-1B, Llama-3.1-3B, and Llama-3.1-8B) to evaluate effectiveness across different model variants as well as the scaling behaviors across different model capacities within our computational constraints for fine-tuning.

For the analysis, we use the normalized mean squared error (NMSE) as in (Shojaee et al., 2025b): $\text{NMSE} = \frac{\sum_{i=1}^{N_{\text{test}}} (\hat{y}_i - y_i)^2}{\sum_{i=1}^{N_{\text{test}}} (y_i - \bar{y})^2}$ on both in-domain (ID) and out-of-domain (OOD) test settings, where $N_{\text{test}}$

---

[2]https://gplearn.readthedocs.io/en/stable/
[3]https://github.com/MilesCranmer/PySR

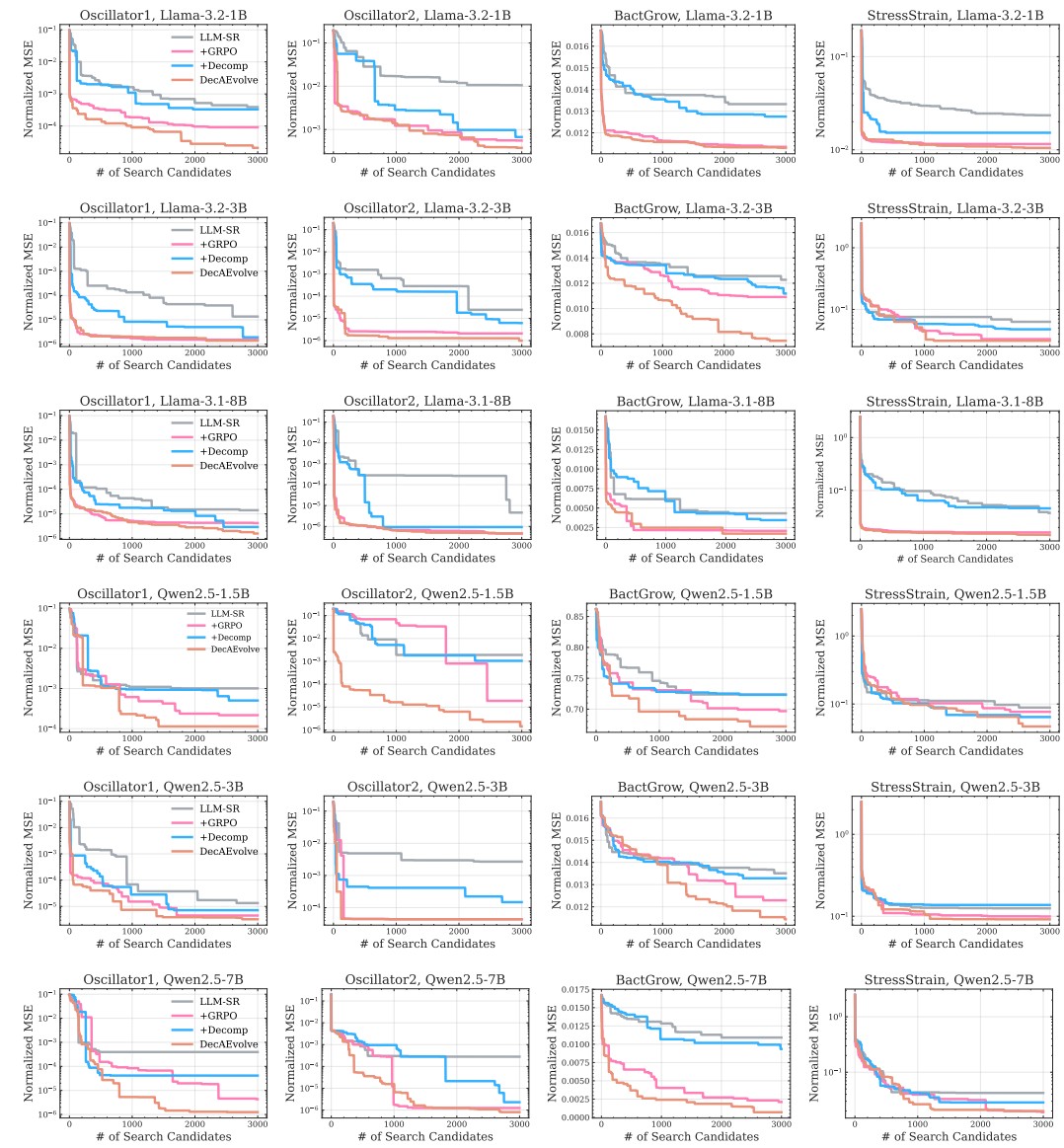

Figure 2: Best-score trajectories of DecAEvolve and its variants against the LLM-SR baseline across benchmark problems. Adaptation (+GRPO) and decomposition (+Decomp) each enhance discovery effectiveness and efficiency, yielding more accurate final equations with fewer search candidates. Their integration in DecAEvolve achieves the best result across all datasets (lower is better). Each curve shows the average across five runs.

is the test size and $\bar{y}$ the mean target value. NMSE normalizes errors by scale of dataset variance, enabling comparison across datasets.

## 4.1 RESULTS

To assess the contribution of our proposed framework DecAEvolve and its key components—decomposition and adaptation—on top of the default evolutionary discovery framework, we compared against the LLM-SR baseline under the same LLM backbones across multiple benchmark datasets. Figure 2 reports the discovery trajectories, showing the progression of the best-achieved normalized MSE (NMSE) across the search process. The results highlight three consistent trends. First, both ablated variants improve over the baseline: the Adaptation (+GRPO) and Decomposition (+Decomp) modules both help to accelerate convergence and lower discovery error. Second, these improvements hold across diverse LLM backbones (Llama-3.2-1B, Llama-3.2-3B, Llama-3.1-8B, Qwen-2.5-1.5B, Qwen-2.5-3B, Qwen-2.5-7B), indicating that the gains are not model-specific but

| Model | Oscillation 1 | | Oscillation 2 | | E. coli growth | | Stress-Strain | |
|---|---|---|---|---|---|---|---|---|
| | ID↓ | OOD↓ | ID↓ | OOD↓ | ID↓ | OOD↓ | ID↓ | OOD↓ |
| GPlearn | 0.0155 | 0.5567 | 0.7551 | 3.188 | 1.081 | 1.039 | 0.1063 | 0.4091 |
| NeSymReS | 0.0047 | 0.5377 | 0.2488 | 0.6472 | N/A ($d > 3$) | | 0.7928 | 0.6377 |
| E2E | 0.0082 | 0.3722 | 0.1401 | 0.1911 | 0.6321 | 1.4467 | 0.2262 | 0.5867 |
| DSR | 0.0087 | 0.2454 | 0.0580 | 0.1945 | 0.9451 | 2.4291 | 0.3326 | 1.108 |
| uDSR | 0.0003 | 0.0007 | 0.0032 | 0.0015 | 0.3322 | 5.4584 | 0.0502 | 0.1761 |
| PySR | 0.0009 | 0.3106 | 0.0002 | 0.0098 | 0.0376 | 1.0141 | 0.0331 | 0.1304 |
| SINDy | 0.9888 | 0.7097 | **4.6e-16** | **1.45e-8** | 1.078 | 1.039 | 0.0781 | 3.5e+15 |
| LLM-SR (Mixtral) | **7.89e-8** | 0.0002 | 0.0030 | 0.0291 | 0.0026 | 0.0037 | 0.0162 | 0.0946 |
| LLM-SR (GPT-3.5-turbo) | 4.65e-7 | 0.0005 | 2.12e-7 | 3.81e-5 | 0.0214 | 0.0264 | 0.0210 | 0.0516 |
| LLM-SR (Llama-3.2-1B) | 0.0003 | 0.1121 | 0.0105 | 0.0543 | 0.0133 | 0.3544 | 0.0934 | 0.3821 |
| LLM-SR (Llama-3.2-3B) | 1.41e-5 | 0.0014 | 0.0021 | 0.0053 | 0.0122 | 0.0588 | 0.0629 | 0.1672 |
| LLM-SR (Llama-3.1-8B) | 1.36e-5 | 0.0009 | 4.61e-6 | 0.0001 | 0.0117 | 0.0240 | 0.0376 | 0.0761 |
| LLM-SR (Qwen2.5-1.5B) | 0.0011 | 0.1233 | 0.0027 | 0.0721 | 0.7237 | 0.9483 | 0.1249 | 0.2435 |
| LLM-SR (Qwen2.5-3B) | 0.0003 | 0.0168 | 0.0018 | 0.0432 | 0.0135 | 0.8011 | 0.0905 | 0.2085 |
| LLM-SR (Qwen2.5-7B) | 1.33e-5 | 0.0017 | 0.0002 | 0.0011 | 0.0109 | 0.1285 | 0.0423 | 0.1851 |
| DecAEvolve (Llama-3.2-1B) | 2.09e-5 | 0.0011 | 0.0018 | 0.0136 | 0.0114 | 0.0698 | 0.0704 | 0.0924 |
| DecAEvolve (Llama-3.2-3B) | 1.57e-6 | 0.0004 | 0.0003 | 0.0005 | 0.0074 | 0.0102 | 0.0311 | 0.0358 |
| DecAEvolve (Llama-3.1-8B) | 1.37e-6 | 0.0002 | 3.64e-7 | 2.11e-5 | 0.0019 | 0.0045 | **0.0144** | **0.0322** |
| DecAEvolve (Qwen2.5-1.5B) | 0.0001 | 0.0784 | 1.22e-6 | 0.0012 | 0.6719 | 0.9211 | 0.0916 | 0.1134 |
| DecAEvolve (Qwen2.5-3B) | 3.23e-6 | 0.0002 | 4.36e-5 | 0.0008 | 0.0115 | 0.0454 | 0.0487 | 0.1612 |
| DecAEvolve (Qwen2.5-7B) | 1.25e-6 | **1.51e-5** | 8.06e-7 | 1.64e-5 | **0.0007** | **0.0012** | 0.0198 | **0.0322** |

Table 1: Comparison of DecAEvolve with SR baseline models on different scientific benchmark problems, measured by Normalized Mean Squared Error (lower is better) over five runs. **The best performance** for each dataset is in bold, and the second best performance is underlined.

instead stem from the principled design of the framework components that transfer across different backbones. Finally, the full DecAEvolve framework, which integrates all three components of evolution, decomposition, and adaptation, consistently delivers the lowest terminal NMSE and the fastest convergence rate in the discovery process.

Table 1 provides a quantitative comparison of DecAEvolve against both non-LLM baselines and the LLM-based baseline LLM-SR across in-domain (ID) and out-of-distribution (OOD) evaluations. We observe that DecAEvolve mostly outperforms state-of-the-art non-LLM methods (e.g., PySR, uDSR, SINDy) as well as the LLM-SR baseline when evaluated under the same LLM backbones. These improvements are mostly robust across both ID and OOD test sets, demonstrating not only higher accuracy but also stronger generalization to unseen data distributions. Performance gains are particularly pronounced with larger backbones such as Llama-3.1-8B and Qwen-2.5-7B, which is expected given that the success of DecAEvolve relies on two key components: (1) decomposition, which requires sufficient reasoning capacity to interpret granular feedback, and (2) adaptation, which depends on reinforcement learning finetuning to exploit reward signals from observed scientific data. Larger models are better able to leverage both of these mechanisms, resulting in consistently stronger performance. Nevertheless, we also find that DecAEvolve with smaller backbones can achieve results that are competitive with, and in some cases better than, the originally reported LLM-SR performance using much larger models such as Mixtral and GPT-3.5. This underscores the critical role of adaptation in scientific discovery: by tailoring even modestly sized open-source models to the specific scientific system, DecAEvolve can surpass the performance of significantly larger general-purpose models.

Lastly, Figure 3 shows consistent reward improvement during GRPO adaptation across both model scales and all datasets, validating our reinforcement learning fine-tuning approach as test-time adaptation for equation discovery. Notably, we observe some scale-dependent behaviors where smaller models show more noise in their RL and reward improvement process than their larger model counterparts. Interestingly, the smaller model usually matches larger model performance eventually even on complex datasets, suggesting that targeted adaptation through GRPO can help to effectively bridge the capability gap between model scales for scientific discovery tasks.

## 5 RELATED WORK

**Symbolic regression and Scientific Discovery.** Early research in symbolic regression and scientific discovery established a foundation for automated equation finding, relying on genetic programming and evolutionary search to explore hypothesis spaces (Koza, 1994a; Cava et al., 2021a). While effective on small problems, these approaches struggled with scalability and tended to rediscover

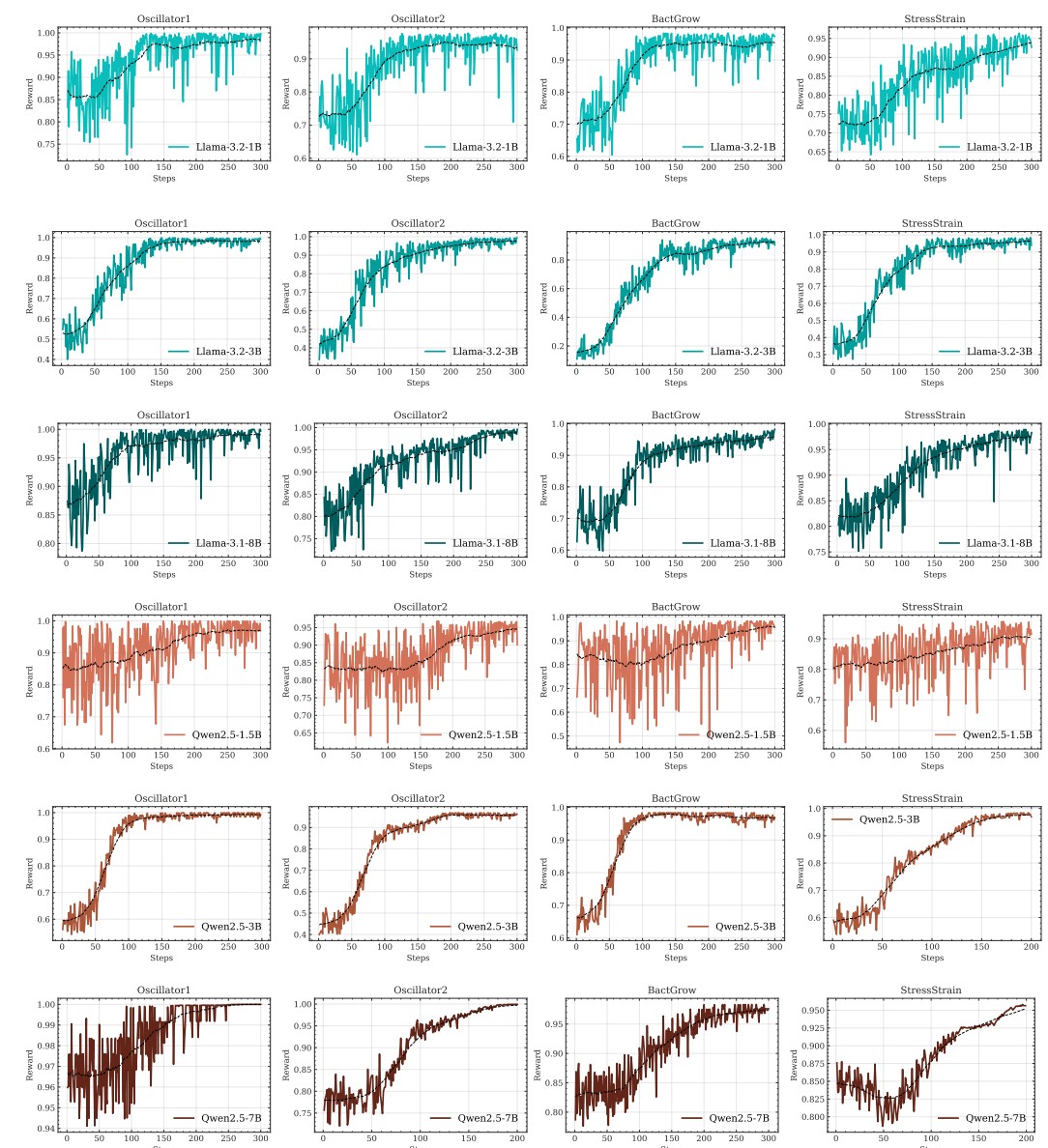

Figure 3: Reward improvements across different models and datasets, showing the success of adaptation with GRPO RL fine-tuning.

shallow functional forms. Later neural-guided methods, such as sparse regression approaches like SINDy (Brunton et al., 2016), AI Feynman (Udrescu & Tegmark, 2020a), physics-inspired constraints (Bruneton, 2025b), and transformer-based symbolic generation methods (Kamienny & colleagues, 2022; Shojaee et al., 2023b; Meidani et al., 2024a) extended these capabilities but often suffered from poor scalability and limited structural diversity. The rise of large language models shifted this landscape. Works such as LLM-SR (Shojaee et al., 2025a) reframed symbolic regression as program synthesis, allowing models to generate equation skeletons enriched by internal scientific priors. Subsequent frameworks expanded this view: LaSR (Grayeli et al., 2024a) guided search with abstracted concepts extracted from prior successes, while bilevel optimizers (Ma et al., 2024a) combined symbolic hypothesis generation with simulation-driven parameter tuning. Benchmarks such as LLM-SRBench (Shojaee et al., 2025c) highlighted both the promise of these methods and their limitations, showing that LLMs, even when coupled with evolutionary refinement, fails to capture the adaptive strategies that real scientific discovery demands.

**Test-time adaptation.** Test-time adaptation has recently emerged as a way to adapt models during inference, guiding models toward novel distributions without additional offline training. In reasoning benchmarks such as ARC-AGI (Chollet et al., 2024), gradient-based test-time training (TTT) has shown great performance in better adapting models to tasks that require more novelty (Akyürek et al., 2024). The ARC-AGI 2024 report similarly attributes recent state-of-the-art results to pipelines that incorporate test-time training components into the problem-solving process (Chollet & Team, 2024). Beyond empirical advances, recent theoretical analyses establish conditions under which a single gradient step at inference provably enhances transformers as in-context learners (Gozeten et al., 2025). Extending beyond supervised updates, Zuo et al. (2025) introduce test-time reinforcement learning (TTRL), where models adapt using consensus-based rewards rather than labels, yielding further improvements across reasoning and math tasks. Despite the successes and potential benefits of test-time training in tasks that require better adapting to novelty, test-time adaptation remains largely unexplored in scientific discovery frameworks. It is still unclear exactly how inference-time learning can align the priors of a model by leveraging the dynamics of specific scientific system during the evolutionary process of search towards discovery.

**Evolution and Prompt Optimization.** A parallel line of work focuses on evolutionary search and optimization of prompts rather than model weights, treating instructions and in-context exemplars as a inference-time search space. Yang et al. (2023) propose OPRO, which frames prompt design as black-box optimization and iteratively improves instructions through feedback with LLMs as optimizer. Guo et al. (2025) extend this perspective with EvoPrompt, combining evolutionary operators such as mutation and crossover with LLMs to explore diverse prompt populations. More recently, Opsahl-Ong et al. (2024) develop MIPRO, a system that jointly optimizes instructions and demonstrations in multi-stage LM programs, demonstrating robust improvements without weight updates. Agrawal et al. (2025) introduce GEPA, which leverages reflective prompt evolution and self-feedback to surpass reinforcement learning baselines like GRPO, achieving higher efficiency in both code and reasoning tasks. Surveys on evolution and prompt optimization synthesize these approaches and position prompt evolution as a label- and compute-efficient alternative to general RL fine-tuning (Ramnath et al., 2025). Our framework builds on this motivation of self-evolving optimization via prompting along with the test-time model adaptation to search deeper and more efficient in the large combinatorial hypothesis spaces of scientific discovery.

# 6 CONCLUSION

We introduce DecAEvolve, a framework that enhances LLM-based equation discovery through granular term-level directional feedbacks, test-time adaptation via GRPO and evolutionary search with LLMs. Our approach transforms static hypothesis generation into adaptive learning, enabling LLMs to progressively align with nuances of underlying observed scientific systems through reinforcement learning model adaptation and interpretable feedback mechanisms. Experimental results across diverse benchmark datasets demonstrate that DecAEvolve consistently outperforms state-of-the-art baselines in both discovery accuracy and search efficiency, while maintaining strong out-of-domain generalization. The success of smaller models through targeted test-time adaptation suggests promising directions for democratizing scientific discovery tools without requiring large, resource-intensive models. Future work could extend our simple decomposition mechanisms to more complex reflection structures and explore better optimization strategies for the evolutionary process. The term-level feedback approach developed here may also prove valuable for systems with highly correlated components and the broader program synthesis tasks requiring iterative refinement in the symbolic space of programs based on component-level understanding.

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

# APPENDIX

## A  LLM-SR MULTI-ISLAND EVOLUTIONARY BUFFER & SAMPLING

LLM-SR (Shojaee et al., 2025a) maintains a multi-island evolutionary experience buffer designed to preserve structural diversity and prevent premature convergence during the process of symbolic equation discovery. The buffer is organized as $\mathcal{P}_t = \bigcup_i \mathcal{P}_t^{(i)}$, where each $\mathcal{P}_t^{(i)}$ is an independently evolving island/population containing pairs of symbolic programs and their corresponding scalar fitness scores. Each island begins with the initial prompt equation $v_0$ with score $s_0$ supplied to the framework at the beginning: $\mathcal{P}_0^{(i)} = \{(v_0, s_0)\}$. Although all islands start identically, they diverge over the search process and evolution. At iteration $t$, the LLM generates a batch of programs $\mathcal{F}_t = \{f_j\}_{j=1}^b$, each associated with a source island, which is the island from which its in-context examples were sampled. Each equation program candidates contains a placeholder vector of parameters which are then optimized with respect to the observed data with the help of BFGS optimizer in python. After parameter optimization, each program receives a fitness score $\text{Score}_{\mathcal{T}}(f, \mathcal{D})$ which is computed as negative mean squared error (MSE) with respect to data:

$$\text{Score}_{\mathcal{T}}(f, \mathcal{D}) = -\frac{1}{n} \sum_{i=1}^n (f(\mathbf{x}_i) - y_i)^2$$

, where $\mathcal{D} = \{(\mathbf{x}_i, y_i)\} \in \mathbb{R}^d \times \mathbb{R}$ refers to observed datapoints. Notably, a program is inserted *only* into its source island if its fitness exceeds that island's current best:

$$\mathcal{P}_t^{(i)} \leftarrow \mathcal{P}_{t-1}^{(i)} \cup \{(f, s)\} \text{ if } s > s_{\text{best}}^{(i)}$$

This ensures that each island progressively specializes in a distinct region of the hypothesis space to avoid local minima and encourage diversity in the search process towards discovery. In LLM-SR, programs are also grouped into clusters within each island using a simple signature corresponding to score $s$ where all programs with identical fitness scores fall into the same cluster. This prevents over-representation of structurally similar programs and maintains a lower-level diversity within each island. After every $T_{\text{reset}} = 4hr$, the algorithm identifies the worst-performing half of the islands as $\mathcal{W} = \arg\min_i s_{\text{best}}^{(i)}$. For each $i \in \mathcal{W}$, the entire island is replaced by a copy of the best-performing equation from a randomly selected surviving island. This mechanism is designed to remove stagnating islands and increases global exploration during the discovery .

Each iteration begins by sampling $k$ equations from the multi-island buffer to construct the few-shot prompt. Sampling follows a hierarchical procedure. First, an island index is sampled uniformly $i \sim \text{Uniform}\{1, \ldots, I\}$. This is mainly to prevent dominant islands from monopolizing the prompt, encouraging cross-island exploration. Within the selected island $P_t^{(i)}$, sampling proceeds in two steps: *Cluster Selection* and *Program Selection*.

For each cluster $c$, we have the mean fitness $s_c = \text{mean}\{s : (f, s) \in c\}$ and clusters are sampled according to Boltzmann weights as:

$$p(c) = \frac{\exp(s_c/\tau_c)}{\sum_{c'} \exp(s_{c'}/\tau_c)}$$

. In this sampling, the temperature $\tau_c$ anneals with island size $u$ as

$$\tau_c = T_0 \left(1 - \frac{u \bmod N}{N}\right)$$

where $T_0 = 0.1$ and $N = 10{,}000$. Within a cluster, programs are sampled with preference for better scores and shorter length. The sampling distribution for each program follows

$$p(f_j) \propto \exp(-\tilde{\ell}_j / \tau_p)$$

where sampling temperature $\tau_p = 1$. Here, $f_j$ refers to the program, $\ell_j$ refers to the corresponding length, and $\tilde{\ell}_j$ is defined as

$$\tilde{\ell}_j = \frac{\ell_j - \min_j \ell_j}{\max_j \ell_j + 10^{-6}}$$

After sampling in-context examples with above procedure, the $k$ sampled programs are serialized into a structured few-shot prompt supplied to the LLM. This prompt acts as the guiding context for the next generation of hypotheses $\mathcal{F}_{t+1}$, completing the LLM-SR evolutionary search and self-reflection mechanism.

## B   DETAILED TERM DECOMPOSITION AND CONTRIBUTION ATTRIBUTION

We give a complete account of how generated programs are decomposed into symbolic terms and how single and pairwise contributions are computed in our implementation. This procedure follows the implementation of evaluator in Shojaee et al. (2025a) with the additional steps: the function body is parsed into an abstract syntax tree (AST), simple assignment chains are inlined, the returned expression is decomposed at additive nodes, and ablation is carried out by rewriting only the final assignment `return` and re-executing in a sandbox. *All annotations are serialized as inline comments in the program without changing executable semantics.* Unless otherwise specified, all ablations include re-optimizing of the remaining parameters on the dataset to ensure that contributions reflect structural differences on instead of suboptimal tuning or parameterization

Operationally, the evaluator isolates the evolved function body, executes it in a sandbox to obtain a scalar score, and then reconstructs the returned expression by expanding intermediate assignments via an assignment map and dependency graph before parsing with Python's `ast` module. From the resulting atoms $\{\tau_t\}$, we perform ablation-based attribution: for each term $t$, we remove $\tau_t$, rebuild a syntactically valid RHS with correct parentheses, re-execute the modified program, and compute a marginal contribution $\Delta_t = \text{Score}_{\mathcal{T}}(f, D) - \text{Score}_{\mathcal{T}}(f_{\setminus t}, D)$, where $S$ is the evaluator score (negative MSE). We analogously compute pairwise signals $\Delta_{t,u}$ by removing $(\tau_t, \tau_u)$. The evaluator writes these results back as inline comments within the function body, so subsequent iterations can consume structured, term-level feedback rather than a single scalar reward. This AST-centric pipeline is lightweight, robust to multi-line programs, and provides the granular guidance that underpins our evolutionary refinement.

**Decomposition to atomic terms**   Let $\hat{y}$ denote the expression returned by the function (or a final assigned variable). We parse the equation program skeleton (body of the function) into an AST, build a line-level assignment map, inline $\hat{y}$ if it is an intermediate variable, and traverse the AST with the following rules: (i) *addition/subtraction* split terms, (ii) *multiplication/division/power* subtrees are preserved as atomic units, and (iii) *unary operators and function calls* (e.g., sin, exp, `np.abs`) are atomic operators. The resulting model has the form $\hat{y} = f(\mathbf{x}; \texttt{params})$ with

**Single-term ablation and contribution.**   After identifying terms from previous step, For each term $u_m$, we form the ablated version of the original equation program as $f_{\setminus u_m}$. After ablation, the remaining equation parameters from placeholder parameter vector `params` ($\texttt{params}_{\setminus u_m}$) are re-optimized on dataset $\mathcal{D}$ (using the same BFGS parameter optimization procedure as in (Shojaee et al., 2025a) with Scipy library in Python). The term contribution is defined as influence function with:

$$\Delta_{u_m} \triangleq \text{Score}_{\mathcal{T}}(f, \mathcal{D}) - \text{Score}_{\mathcal{T}}(f_{\setminus u_m}, \mathcal{D}). \tag{2}$$

If removal yields invalid outputs, we treat term $u_m$ as essential and assign maximal contribution under the current score scale.

**Pairwise interaction.**   Similarl to the single-term contribution score estimation, for a pair $(u_m, u_n)$ we define ablated function as $f_{\setminus \{u_m, u_n\}}$ and the corresponding contribution score for this pair-wise

```python
def equation(x, v, params):
    """ Mathematical function for acceleration in damped nonlinear oscillator """

    # Individual Term Contributions:
    # [Ablation] Removing this term decreases the score by 0.00394026: params[0]*x
    # [Ablation] Removing this term decreases the score by 0.00000446: params[2]
    # [Ablation] Removing this term decreases the score by 0.00000001: params[1]*v

    # Term Pair Contributions:
    # [Ablation] Removing these terms together decreases the score by 0.00414153:
    #   Term 1: params[0]*x
    #   Term 2: params[2]
    # [Ablation] Removing these terms together decreases the score by 0.00394474:
    #   Term 1: params[0]*x
    #   Term 2: params[1]*v
    # [Ablation] Removing these terms together decreases the score by 0.00000450:
    #   Term 1: params[1]*v
    #   Term 2: params[2]

    return params[0] * x + params[1] * v + params[2]
```

Figure 4: **A simple example of program-level annotations.** Candidate equation for a damped nonlinear oscillator (simple linear here) is annotated in-line with single-term and pairwise ablation contributions (as comments) immediately above the `return` statement. The evaluator computes these contribution scores after re-optimizing remaining parameters via BFGS, decomposing the return expressions, and re-evaluating ablations in a sandbox.

interaction subtree as:

$$\Delta_{u_m, u_n} \triangleq \text{Score}_{\mathcal{T}}(f, \mathcal{D}) - \text{Score}_{\mathcal{T}}(f_{\setminus \{u_m, u_n\}}, \mathcal{D}). \tag{3}$$

As with the single-term case, the remaining parameters after parwise term ablation are re-optimized after removal to isolate the structural interaction effect without suboptimal parameterization. These values reveal redundancy versus synergy by comparing $\Delta_{u_m, u_n}$ against the sing-term counterparts $\Delta_{u_m}$ and $\Delta_{u_n}$.

**Annotation and persistence.** After computing $\{\Delta_{u_m}\}$ and $\{\Delta_{u_m, u_n}\}$, we serialize them as human-readable inline comments directly in the equation skeleton programming function body (above the `return` statement), and store the annotated program in the experience buffer. This preserves executable semantics while exposing interpretable, decomposition-based directional feedback that guides subsequent process of discovery towards better sub-terms in the hypothesis space.

## B.1 AST-BASED DECOMPOSITION

An Abstract Syntax Tree (AST) is a language-agnostic representation of a program that makes explicit the hierarchical composition of an expression. Internal nodes correspond to operators or function applications (e.g., `+`, `*`, `**`, `np.sin`), and leaves correspond to parameters or input variables. In our setting, we parse each LLM-generated equation skeleton Python program into an AST and split at additive nodes, while preserving multiplicative, divisional, power, and functional subtrees as atomic terms. This yields a linear combination $f(x) = \sum_{m=1}^{M} u_m(x)$ of symbolic atoms $u_m$. For example, Figure 5 illustrates this mapping from the generated hypothesis function (left) to its AST (right): a return expression such as $y = t_1 + t_2 + t_3 - t_4$ becomes a top-level sum where each $t_i$ is an intact subtree, enabling principled symbolic decomposition without altering operator precedence.

## C DETAILED TEST-TIME ADAPTATION WITH REINFORCEMENT LEARNING

We formulate our test-time training/adaptation procedure as reinforcement learning over a deterministic Markov Decision Process (MDP) $\mathcal{M} = (S, A, R, T)$.

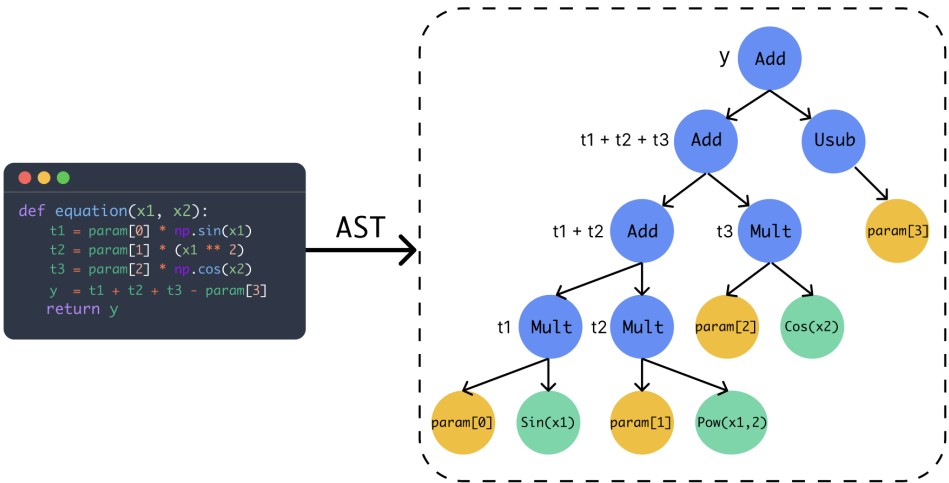

Figure 5: Parsing an equation program into an AST.

**States.** Similar to the reinforcement learning LLM fine-tuning, the state space $S$ in this setting consists of partial sequences $(p, h_{1:k})$ where $p$ is the prompt and $h_{1:t}$ is a prefix of the generated output until token $t$.

**Actions.** At each step of this reinforcement learning test-time adaptation, the action space $A$ corresponds to the next token from vocabulary $h_{t+1} \in V$.

**Rewards.** In our setting, rewards are assigned based on the execution of the equation program on the observed data and only obtained at terminal states. For a completed program $h$, we execute the synthesized program on the validation set and compute the reward as noted in Section 3.1 with $r(p, h) = \text{Score}_{\mathcal{T}}(h, \mathcal{D}) = \exp(-\text{MSE}(h, \mathcal{D}))$ with invalid completions included in training and assigned a fixed floor reward of $r = 0.01$.

**Implementation details** After generating a new batch of candidate equation hypotheses and evaluating their performance, the model updates its policy through GRPO (Shao et al., 2024). At iteration $n$, we treat the prompt $\mathbf{p_n}$ together with its group of completions $h_{i\ i=1}^{n\ G}$ and their associated rewards $\{r_i^n\}_{i=1}^{G}$ as a single training step. These prompt–completion groups accumulate over iterations and serve as the training data for GRPO updates. We fine-tune using Adam with learning rate $10^{-6}$ and a warmup–stable–decay schedule (200 warmup steps). Training uses an effective batch size of 64 (16 per device with gradient accumulation 4). Each prompt is sampled with $G = 64$ completions at temperature $0.8$ and top-$p = 0.9$. Only LoRA adapter parameters are updated ($r = 16$, $\alpha = 16$, dropout 0.05), while the base model remains frozen as $\pi_{\text{ref}}$.

# D HOW IS DECOMPOSITION DIFFERENT THAN MOOSE-CHEM3?

One of the relevant works from literature with the same high-level motivation of incorporating decomposition in the scientific discovery is MOOSE-Chem3 (Liu et al., 2025). MOOSE-Chem3 targets the discovery and ranking of experimental chemistry hypotheses. Each hypothesis is a natural-language description of a reaction mechanism or material design strategy. Their decomposition procedure is mechanistic: the framework identifies functional chemical components (e.g., polymer matrices, redox pairs, electrode structures, etc.) to build a qualitative similarity measure for experiment-guided hypothesis ranking. This process is domain-specific, mechanistic, and grounded in wet-lab feedback or its simulated analogue. In contrast, DecAEvolve focuses on scientific equation discovery, where hypotheses are explicit symbolic expressions. Our decomposition is quantitative and analytic: an equation is ablated into symbolic terms and operators, and an influence function is used to measure each component's contribution to prediction error and structural behavior with respect to data. These influence signals directly shape mutation choices, and the structured feedback used during evolution.

Although both frameworks use the general idea of breaking hypotheses into components to find successful components, the decomposition differ fundamentally in (i) the objects being decomposed (mechanistic chemical descriptions vs. symbolic math relations), (ii) the nature of decomposition (qualitative functional roles vs. quantitative influence), and (iii) the role of decomposition within the discovery framework (similarity-based ranking vs. fine-grained feedback for optimization). Moreover, decomposition is only one part of DecAEvolve that consists of a hybrid three-module system (Decompose, Adapt, Evolve), where term-level analysis interacts directly with RL-based test-time adaptation, an approach not explored in MOOSE-Chem3 or prior discovery methods.

## E    BASELINE IMPLEMENTATION DETAILS

We evaluate DecAEvolve against the baseline methods reported in Shojaee et al. (2025a), including GPlearn, PySR, DSR, uDSR, NeSymReS, and E2E. These baselines represent diverse SR approaches spanning genetic programming, reinforcement learning, and pre-trained transformers. For implementation details and hyperparameters of these methods, we refer readers to Appendix A of Shojaee et al. (2025a). Below, we provide additional details for the SINDy baseline, which we newly evaluate in this work.

## F    ADDITIONAL EXPERIMENTS

### F.1    COMPARISON WITH SINDY BASELINE

We evaluate SINDy: Sparse Identification of Nonlinear Dynamics method (Brunton et al., 2016) as an additional non-LLM baseline using the PySINDy implementation (Kaptanoglu et al., 2022). SINDy discovers governing equations by performing sparse regression over a predefined library of candidate functions, assuming dynamics can be expressed as sparse linear combinations of nonlinear basis functions.

**Experimental Setup.**    We configured SINDy with a polynomial library (degree 3) augmented with Fourier terms (frequencies up to 2) to provide a balanced function library without excessive expansion. The STLSQ optimizer used sparsity threshold 0.1, regularization $\alpha = 0.01$, and 20 maximum iterations.

| Model | Oscillation 1 | | Oscillation 2 | | E. coli growth | | Stress-Strain | |
|---|---|---|---|---|---|---|---|---|
| | ID↓ | OOD↓ | ID↓ | OOD↓ | ID↓ | OOD↓ | ID↓ | OOD↓ |
| PySR | 0.0009 | 0.3106 | 0.0002 | 0.0098 | 0.0376 | 1.0141 | 0.0331 | 0.1304 |
| SINDy | 0.9888 | 0.7097 | **4.62e-16** | **1.45e-8** | 1.078 | 1.039 | 0.0781 | 3.52e+15 |
| LLM-SR (Qwen2.5-7B) | 1.33e-5 | 0.0017 | 0.0002 | 0.0011 | 0.0109 | 0.1285 | 0.0423 | 0.1851 |
| DecAEvolve (Qwen2.5-7B) | **1.25e-6** | **1.51e-5** | 8.06e-7 | 1.64e-5 | **0.0007** | **0.0012** | **0.0198** | **0.0322** |

Table 2: Comparison of SINDy with representative SR baselines (best non-LLM method PySR, LLM-based LLM-SR, and our DecAEvolve) using Qwen2.5-7B backbone.

**Results and Analysis.**    Table 2 shows SINDy achieves the best performance on Oscillation 2 (NMSE $< 10^{-7}$ on OOD) among all baselines. This is because Oscillation 2 represents a typical dynamical system, the type SINDy was explicitly designed for, where dynamics follow a linear form with nonlinear basis functions (polynomials and trigonometrics). Given an appropriate library containing the true functional forms, SINDy is highly efficient for such problems. However, SINDy exhibits severe limitations on problems outside its linear-formulation assumption. On Bacterial Growth, it completely fails (NMSE $\approx 1.08$) as the ground-truth involves products of multiple nonlinear terms, which cannot be represented as linear combinations. On Stress-Strain, while fitting training data reasonably (NMSE $= 0.078$), it suffers catastrophic extrapolation failure on OOD data (NMSE $= 3.52 \times 10^{15}$) as polynomial approximations diverge outside the training range. On Oscillation 1, performance degrades significantly (NMSE $= 0.99$ ID) when dynamics deviate from SINDy's assumed form, demonstrating sensitivity to equation structure even within the dynamical systems

domain. These results highlight SINDy's fundamental constraint: performance is entirely determined by whether the true equation lies within the predefined library's representational capacity.

## F.2 ADDITIONAL SAMPLE BUDGET FOR INFERENCE FRAMEWORKS

We also conducted additional experiments to run inference search variants (LLM-SR and +Decomp) for total number of samples used by the GRPO-based variants (DecAEvolve and +GRPO). Specifically, we run LLM-SR and +Decomp for additional 12800 samples ($64 \times 200$) with Qwen2.5-7B model backbone. As it can be observed from Figure 2 and Figure 6, the performacne of LLM-SR and +Decomp is already well-converged by roughly 3000 samples, and allocating an additional 12800 samples offers no meaningful performance improvement. This result shows that DecAEvolve leverages these samples more effectively for discovery than baselines.

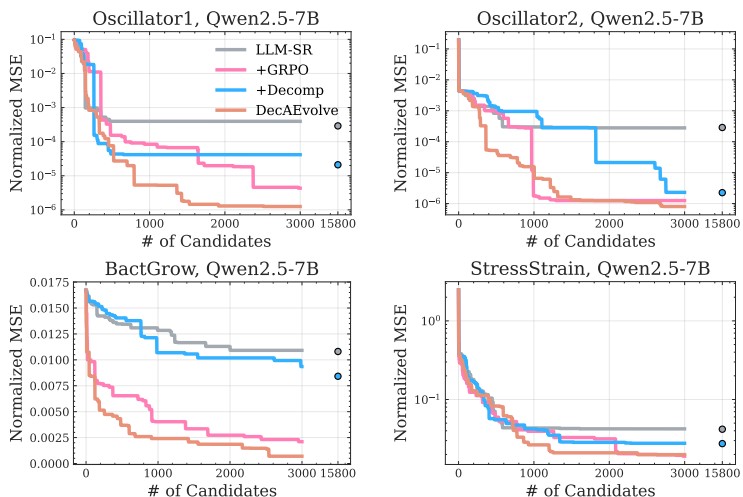

Figure 6: The impact of additional GRPO-matched samples on inference algorithms.

## F.3 IMPACT OF PARAMETER RE-OPTIMIZATION IN DECOMPOSITION

We conducted additional ablation study to compare the impact of two strategies of parameter optimization during the decomposition with structure ablations: (1) *Re-opt Parameters*: After ablating a decomposed sub-term, we re-optimize the remaining parameters to obtain the best fit for the modified structure; and (2) *Freeze Parameters*: After ablating a term, we keep the remaining parameters fixed to the values learned from the original full structure of equation. Symbolic terms of an equation are often highly coupled, and modifying part of the structure can shift the optimal values of the remaining parameters. This coupling is indeed a general challenge across all equation discovery methods. However, our goal is to attribute performance changes to structural differences, not to artifacts of suboptimal parameterization. Re-optimizing parameters ensures that each ablated structure is evaluated at its own best performance—providing a more faithful estimate of the true contribution of each symbolic component. To examine this empirically, we performed a focused ablation study using the Qwen2.5-7B backbone across several representative datasets (Oscillator1, Oscillator2, BactGrow, and StressStrain). Results are shown in Figure 7. The "Re-opt Params" curves consistently achieve lower normalized MSE than the "Freeze Params" curves, particularly in datasets with strong nonlinear coupling such as BactGrow and StressStrain. This indicates that re-optimization yields more reliable and stable assessments of structural contributions.

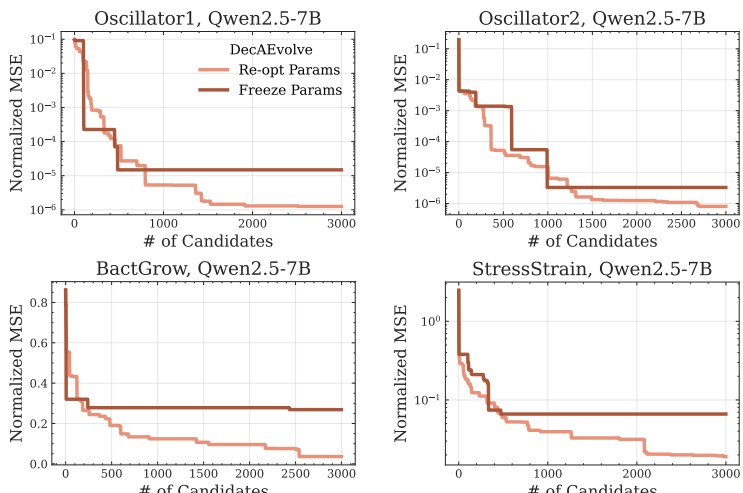

Figure 7: Ablation results comparing DecAEvolve decomposition with and without re-optimizing the parameters after structure ablation.

## G    INITIAL INPUT PROMPTS

The prompts in Figs. 8–11 were used for evaluating DecAEvolve on the four regression tasks from the LLM-SR(Shojaee et al., 2025a) benchmark. Each prompt specifies the target problem and a Python function template with placeholder parameters. In all cases, the prompts shown below correspond to the initial call to the LLM. In subsequent iterations, DecAEvolve augments the prompt with examples drawn from top-performing hypotheses in the evolving buffer, enabling in-context learning from previously discovered candidates and their annotated contributions.

```
You are a helpful assistant tasked with discovering mathematical function structures for scientific
systems. You are given an example of the function signature in the first function equation_v0 below.
Your task is to complete the last 'equation' function with your mathematical relationship, considering
the physical meaning and relationships of inputs. Only give me the completion code of the BODY of the
current 'equation' FUNCTION. Do NOT give me a new function. Do NOT give me 'pass' instead of
completion. Just give me the new mathematical relationship with inputs for completion of current
function body. Do NOT use equation_v0 in your completion.

"""
Find the mathematical function skeleton that represents acceleration in a damped nonlinear oscillator
system with driving force, given data on position, and velocity.
"""

import numpy as np

#Initialize parameters
MAX_NPARAMS = 10
params = [1.0]*MAX_NPARAMS

def equation_v0(x: np.ndarray, v: np.ndarray, params: np.ndarray) -> np.ndarray:
    """Initial example of equation."""
    dv = params[0] * x  +  params[1] * v  + params[2]
    return dv

def equation_v1(x: np.ndarray, v: np.ndarray, params: np.ndarray) -> np.ndarray:
    """Improved version of `equation_v0`."""
```

Figure 8: Oscillator I input prompt used for evaluating DecAEvolve.

```
You are a helpful assistant tasked with discovering mathematical function structures for scientific
systems. You are given an example of the function signature in the first function equation_v0 below.
Your task is to complete the last 'equation' function with your mathematical relationship, considering
the physical meaning and relationships of inputs. Only give me the completion code of the BODY of the
current 'equation' FUNCTION. Do NOT give me a new function. Do NOT give me 'pass' instead of
completion. Just give me the new mathematical relationship with inputs for completion of current
function body. Do NOT use equation_v0 in your completion.

"""
Find the mathematical function skeleton that represents acceleration in a damped nonlinear oscillator
system with driving force, given data on time, position, and velocity.
"""

import numpy as np

#Initialize parameters
MAX_NPARAMS = 10
PRAMS_INIT = [1.0]*MAX_NPARAMS

def equation_v0(t: np.ndarray, x: np.ndarray, v: np.ndarray, params: np.ndarray) -> np.ndarray:
    """Initial example of equation."""
    dv = params[0] * t + params[1] * x  +  params[2] * v + params[3]
    return dv

def equation_v1(t: np.ndarray, x: np.ndarray, v: np.ndarray, params: np.ndarray) -> np.ndarray:
    """Improved version of `equation_v0`."""
```

Figure 9: Oscillator II input prompt used for evaluating DecAEvolve.

```
You are a helpful assistant tasked with discovering mathematical function structures for scientific
systems. You are given an example of the function signature in the first function equation_v0 below.
Your task is to complete the last 'equation' function with your mathematical relationship, considering
the physical meaning and relationships of inputs. Only give me the completion code of the BODY of the
current 'equation' FUNCTION. Do NOT give me a new function. Do NOT give me 'pass' instead of
completion. Just give me the new mathematical relationship with inputs for completion of current
function body. Do NOT use equation_v0 in your completion.

"""
Find the mathematical function skeleton that represents E. Coli bacterial growth rate, given data on
population density, substrate concentration, temperature, and pH level.
"""

import numpy as np

#Initialize parameters
MAX_NPARAMS = 10
PRAMS_INIT = [1.0]*MAX_NPARAMS

def equation_v0(b: np.ndarray, s: np.ndarray, temp: np.ndarray, pH: np.ndarray, params: np.ndarray) ->
np.ndarray:
    """Initial example of equation."""
    return params[0] * b + params[1] * s + params[2] * temp + params[3] * pH + params[4]

def equation_v1(b: np.ndarray, s: np.ndarray, temp: np.ndarray, pH: np.ndarray, params: np.ndarray) ->
np.ndarray:
    """Improved version of `equation_v0`."""
```

Figure 10: BactGrow input prompt used for evaluating DecAEvolve.

```
You are a helpful assistant tasked with discovering mathematical function structures for scientific
systems. You are given an example of the function signature in the first function equation_v0 below.
Your task is to complete the last 'equation' function with your mathematical relationship, considering
the physical meaning and relationships of inputs. Only give me the completion code of the BODY of the
current 'equation' FUNCTION. Do NOT give me a new function. Do NOT give me 'pass' instead of
completion. Just give me the new mathematical relationship with inputs for completion of current
function body. Do NOT use equation_v0 in your completion.

"""
Find the mathematical function skeleton that represents stress, given data on strain and temperature in
an Aluminium rod for both elastic and plastic regions.
"""

import numpy as np

#Initialize parameters
MAX_NPARAMS = 10
params = [1.0]*MAX_NPARAMS

def equation_v0(strain: np.ndarray, temp: np.ndarray, params: np.ndarray) -> np.ndarray:
    """Initial example of equation."""
    return params[0] * strain  +  params[1] * temp

def equation_v1(strain: np.ndarray, temp: np.ndarray, params: np.ndarray) -> np.ndarray:
    """Improved version of `equation_v0`."""
```

Figure 11: StressStrain input prompt used for evaluating DecAEvolve.

