# OpenReview forum: "DecAEvolve: Decompose, Adapt, and Evolve, or Three Pillars of Effective LLM-based Scientific Equation Discovery"
_ICLR.cc/2026/Conference — Submitted to ICLR 2026_

### Official Review · Reviewer_hhwi · 2025-10-31

**Soundness:** 2
**Presentation:** 2
**Contribution:** 2
**Rating:** 2
**Confidence:** 5

**Summary:**

The paper proposes DecAEvolve, a framework that integrates term-level decomposition, test-time adaptation via reinforcement learning, and evolutionary search, enabling large language models to discover scientific equations more efficiently and interpretably from data.

**Strengths:**

Proposes a unified framework (DecAEvolve) that combines decomposition, test-time adaptation, and evolutionary search in a coherent pipeline.

**Weaknesses:**

My concerns are as follows:
1. The evaluation of each decomposed term is conceptually problematic
1.1 Because the process still relies on refitting parameters using BFGS, the results are prone to overfitting and instability. Each refitting step changes the weights of the remaining terms and may converge to a local optimum, thereby failing to reflect the true importance of the removed term.
1.2 Moreover, the symbolic terms in an equation are often highly coupled and possess domain-specific semantic interdependencies. Removing a single term in isolation can damage interpretability and cannot faithfully represent the true contribution of that component.
1.3 In addition, the computational cost is extremely high. If BFGS optimization must be executed after every single term removal—and given that BFGS itself is relatively slow—this design imposes a substantial computational burden.
In summary, while the method aims to provide granular feedback, I believe such feedback may become distorted due to numerical instability and structural coupling. Furthermore, I did not find clear ablation or reverse-validation experiments demonstrating the actual utility of this design (please correct me if I missed such evidence).

2. As shown in Fig. 1, the proposed framework remains highly similar to the LLM-SR architecture (including the island-based evolutionary design and overall search pipeline). The main difference lies in the addition of a modified GRPO fine-tuning step. While this is not to say that LLM-SR is inadequate, the paper does not introduce a genuinely novel perspective on how LLMs can be applied to equation discovery. The LLM still functions primarily as a solver, and this role does not represent a breakthrough compared with traditional RL or GP approaches.

3. Experimental setup concerns
The experiments are conducted on the LLM-SR proposed dataset, yet the Qwen2.5 models were released after LLM-SR, raising a potential data leakage issue. To ensure fair comparison, the authors should either use the newer LLM-SRBench benchmark or restrict evaluation to models that precede or coincide with the LLM-SR release.
Overall, the LLM-SR dataset is not a mature benchmark—its authors quickly released the improved LLM-SRBench afterward. Therefore, I strongly suggest conducting evaluations on LLM-SRBench instead of the outdated LLM-SR.

**Questions:**

Please refer to the weakness.

---

> ### Author Response · Authors · 2025-11-27
> **Response to Reviewer hhwi (1)**
>
> Thank you for dedicating your time and expertise to review our submission. Please find our responses below.
>
> > The evaluation of each decomposed term is conceptually problematic
> > * 1.1 Because the process still relies on refitting parameters using BFGS, the results are prone to overfitting and instability. Each refitting step changes the weights of the remaining terms and may converge to a local optimum, thereby failing to reflect the true importance of the removed term.
> > * 1.2 Moreover, the symbolic terms in an equation are often highly coupled and possess domain-specific semantic interdependencies. Removing a single term in isolation can damage interpretability and cannot faithfully represent the true contribution of that component.
> > * 1.3 In addition, the computational cost is extremely high. If BFGS optimization must be executed after every single term removal—and given that BFGS itself is relatively slow—this design imposes a substantial computational burden.
> > * In summary, while the method aims to provide granular feedback, I believe such feedback may become distorted due to numerical instability and structural coupling. Furthermore, I did not find clear ablation or reverse-validation experiments demonstrating the actual utility of this design (please correct me if I missed such evidence).
>
> Thank you for the thoughtful comments. We understand the reviewer concern on this, however, we still think that estimating contribution of decomposed terms after re-optimization of remaining parameters is better than using the original values of these parameters which are only based on the original function structure with respect to data. By this comparison of two structures both under their best performance (with their own optimized parameters), we make sure that estimated contributions are due to the structure ablations not the suboptimal tuning and parameterization. As reviewer noted, symbolic terms in an equation are often highly coupled and this is in fact a general challenge in all the equation discovery techniques, however, from empirical results we observe that even in cases with highly coupled non-linear interactions (like in bactrgrow and stress-strain datasets), the simple term decomposition is empirically providing gains in performance. In response to the reviewer's comment, we have conducted ablation experiments with decomposition based on originally optimized parameters instead of the re-optimized ones. Due to the limited time of rebuttal, the experiments are only conducted on Qwen2.5-7B model backbone We have provided the results in Figure 7 of Appendix F.3. From the primary results it can be observed that decomposition with re-optimization of parameters in the ablated structures provides better performance than using parameters from the initial original function structure. We do plan to include a more comprehensive experiments on this ablation across different datasets in the camera-ready version of the paper.
>
> > As shown in Fig. 1, the proposed framework remains highly similar to the LLM-SR architecture (including the island-based evolutionary design and overall search pipeline). The main difference lies in the addition of a modified GRPO fine-tuning step. While this is not to say that LLM-SR is inadequate, the paper does not introduce a genuinely novel perspective on how LLMs can be applied to equation discovery. The LLM still functions primarily as a solver, and this role does not represent a breakthrough compared with traditional RL or GP approaches.
>
> We respectfully disagree with the reviewer. Our contribution goes well beyond just “adding GRPO to LLM-SR.” The core novelty is a principled integration of decomposition, online adaptation, and evolutionary search that directly targets fundamental limitations in current LLM-based equation discovery: (1) inability to adapt to new scientific systems beyond the model’s internal priors, and (2) lack of structural guidance toward promising regions of the hypothesis space. This transforms the LLM from a static candidate proposer into an adaptive learner that incorporates system-specific patterns and structures during search towards discovery — a conceptual shift from prior methods that treat the LLM purely as a solver. The consistent empirical gains over baseline also confirm that DecAEvolve addresses these key limitations.

---

> ### Author Response · Authors · 2025-11-27
> **Response to Reviewer hhwi (2)**
>
> > Experimental setup concerns The experiments are conducted on the LLM-SR proposed dataset, yet the Qwen2.5 models were released after LLM-SR, raising a potential data leakage issue. To ensure fair comparison, the authors should either use the newer LLM-SRBench benchmark or restrict evaluation to models that precede or coincide with the LLM-SR release. Overall, the LLM-SR dataset is not a mature benchmark—its authors quickly released the improved LLM-SRBench afterward. Therefore, I strongly suggest conducting evaluations on LLM-SRBench instead of the outdated LLM-SR.
>
> Thank you for the comment. We believe there is a fundamental distinction between the LLM-SR datasets and the data leakage concern observed in LLM-SR for Feynman benchmark problems. Unlike Feynman problems, which are well-known physics laws with direct 1-1 mappings from textual descriptions to ground-truth equations, the LLM-SR datasets incorporate synthetic components into functions which are imposed to numeric data as the main factor guiding the discovery process. This construction prevents the straightforward prompt-to-equation mapping that causes memorization.
>
> Our empirical evidence further supports this distinction. By looking into LLM-SR discovery curves with Qwen2.5 models in Figure 2, we observe that discovery pattern mostly has a gradual error reduction over iterations instead of a sharp drop at the beginning (~50 iterations) which LLM-SR paper previously observed for Feynman problems as a sign of memorization (Figure 11 in [1]). We also do not see very different discovery patterns in Figure 2 between recent Qwen2.5 models and older models Llama3.1 which could again suggest that leakage might not be a confounding factor in these experiments.
>
> Also, we would like to note that since DecAEvolve requires test-time LLM adaptation/finetuning for each dataset, it requires more computational resources than the existing inference-based discovery approaches like LLM-SR. Evaluation across 239 LLM-SRBench datasets would be computationally very expensive and not possible given our academic resource constraints at the moment. In response to the reviewer's comment, we have started conducting additional experiments during the rebuttal period on LSR-Synth chemistry category from LLM-SRBench. The experiments are still ongoing and we do plan to include the results once experiments are completed in the camera-ready version of paper.
>
> [1] LLM-SR: Scientific Equation Discovery via Programming with Large Language Models, ICLR 2025
>
>
> ---
> We hope that our rebuttal address the reviewer's concerns, and if so, they would consider updating their score. We’d be more than happy to engage in further discussions.

---

### Official Review · Reviewer_aWkA · 2025-10-31

**Soundness:** 2
**Presentation:** 3
**Contribution:** 3
**Rating:** 4
**Confidence:** 3

**Summary:**

# Review for DecAEvolve

## Summary
- The paper proposes a new LLM-based framework for scientific discovery called DecAEvolve. They explain the relevant context and background of other methods and show their proposed framework outperforms others. Their innovation is using test-time RL using GRPO and term decomposition to improve the generated results while iterating on scientific discovery.

## Recommendation
- Weak reject. I actually like the core idea here but there are lots of issues in the paper and experimentation that need to be ironed out. If these issues get addressed then I would be happy to recommend this paper for acceptance. I outline these issues in Questions and Feedback.

**Strengths:**

## Strengths
- The model architecture seems to be novel and results substantiate the design decisions.
- Lots of experiments are performed, including ablation studies, comparison with a similar LLM architecture, and non-LLM models.
- Figures and tables are legible and provide useful information.

**Weaknesses:**

## Weaknesses
- Not enough explanation/details for the proposed method, even in the appendix.
- Key experimentation details are missing/ not substantiated.
- No code is provided to verify the implementation and method is doing what is claimed.

**Questions:**

## Questions
- Where is the code? The method is not proprietary and to properly confirm the results provided here are not fabricated it is best practice to provide *reproducable* code. I should be able to look at and run everything that is claimed in the paper.
- "internal prior" and "scientific prior" are used a lot in the paper, but is never explained. Exactly what is this? Please include this description in the background section.
- For test-time GRPO, how is Score_T computed? The appendix B reference doesn't clarify this at all. It's mentioned there is a held-out test suite, but this test suite is not defined or made clear.
- What is the difference in computation and runtime for your model compared to others? I would imagine it runs slower due to directional feedback (section 3.2) as well as test-time GRPO.
- Why was SINDy not used as a non-LLM baseline? Why was it not mentioned in the paper at all?
- The experiment comparing with LLM-SR is not clearly stated. On one hand, it's mentioned in the main text (line 255) that 3,000 LLM calls were made. On the other hand, Figure 2 shows that there were 3,000 search candidates, implying 3,000 full loops of the method. My questions are:
   - If you used 3,000 loops per problem, what is the runtime comparison between LLM-SR and DecAEvolve per problem? I would expect DecAEvolve to take longer to run due to test-time model tuning using LoRA.
   - If you used 3,000 LLM calls per problem, did you count the LLM calls in GRPO (N times G=64 completions) for DecAEvolve? If so, then DecAEvolve should've had less total loops than LLM-SR due to DecAEvolve's test-time GRPO.

## Feedback
- Please add a citation to and briefly mention how SINDy fits in to the previous work section for neural-guided deep learning methods for discovering symbolic regression:
```
@article{brunton2016discovering,
  title={Discovering governing equations from data by sparse identification of nonlinear dynamical systems},
  author={Brunton, Steven L and Proctor, Joshua L and Kutz, J Nathan},
  journal={Proceedings of the national academy of sciences},
  volume={113},
  number={15},
  pages={3932--3937},
  year={2016},
  publisher={National Academy of Sciences}
}
```
- "Line 92, "and SGA": make this the start of a new sentence, otherwise you have one sentence spanning 6 lines.
- The first sentence of your key insight should be modified to "Our key insight is that equation discovery benefits from both adaptation (aligning the model with data distributions) and decomposition (understanding which symbolic components matter), neither of which prior LLM frameworks integrate. "
  - Mention that LLM frameworks don't do this explicitly, the above citation for SINDy does this exactly just without an LLM.
- For "Problem Formulation" please change $x_i$ to either have a bold $\mathbf{x}_i$ or a vector on top of the $\vec{x}_i$ to denote that it is in $\mathbb{R}^d$ and not just $\mathbb{R}$, since it currently looks identical to $y_i$.
  - The LLM-SR paper chose a bold $\mathbf{x}_i$.
- Algorithm 1 should have another loop in it, since you do Stage 1 and Stage 2 multiple times to get the final solution.
- Algorithm 1 is not clearly explained. What is $E$ and $e_j$? What is $SampleExp$? What about $MakeFewShotPrompt$? All of this needs to be rewritten or explained better. The point of the algorithm is to make it clear what your model is doing.
- Table 1, fix lowest score for column 1 to be LLM-SR (Mixtral) and second best score to be LLM-SR (GPT-3.5).
- It's weird to have a related work section that is very similar to what you have in your introduction. Consider consolidating the sections together and moving the related work section up to the introduction.
- Change line 439/440 "Zue et al." to an in-text citation `\citet{...}` and remove the end-of sentence citation on line 441/442.
  - Same problem end of line 447.
  - Same problem in line 449.
  - Same problem in line 451.
  - Same problem in line 453.
- Line 441, there is a run-on slightly incoherent sentence:
   - Change
     - "Despite these successes and potential benefits of test-time training in better adapting to novelty, test-time adaptation remains largely unexplored in evolutionary scientific discovery frameworks, leaving open how inference-time learning can directly align priors of pretrained model with the dynamics of specific scientific system during the evolutionary process of search and discovery."
   - to
      - "Despite the successes and potential benefits of test-time training in better adapting to novelty, test-time adaptation remains largely unexplored in evolutionary scientific discovery frameworks. It is still unclear exactly how inference-time learning can directly align the priors of a pretrained model by leveraging the dynamics of specific scientific system during the evolutionary process of search and discovery."
   - or something similar.
- Line 465: change "GRPO and, evolutionary" to "GRPO and evolutionary

---

> ### Author Response · Authors · 2025-11-27
> **Response to Reviewer aWkA (1)**
>
> > Weak reject. I actually like the core idea here but there are lots of issues in the paper and experimentation that need to be ironed out. If these issues get addressed then I would be happy to recommend this paper for acceptance. I outline these issues in Questions and Feedback.
>
> Thank you for dedicating your time and expertise to review our submission. We are grateful for the constructive comments and have addressed each of the raised concerns in detail below.
>
> > * Where is the code? The method is not proprietary and to properly confirm the results provided here are not fabricated it is best practice to provide reproducable code. I should be able to look at and run everything that is claimed in the paper.
> > * No code is provided to verify the implementation and method is doing what is claimed.
>
> Here is the anonymous link to the code for this project (https://anonymous.4open.science/r/decaevolve-1215) and we do plan to also publicly open-source the code upon paper acceptance.
>
> > For test-time GRPO, how is Score_T computed? The appendix B reference doesn't clarify this at all. It's mentioned there is a held-out test suite, but this test suite is not defined or made clear.
>
> Thank you for the comment. We have added clarification and more details on this to the updated version of paper (Sec 3.1, L188-190)
>
> > "internal prior" and "scientific prior" are used a lot in the paper, but is never explained. Exactly what is this? Please include this description in the background section.
>
> Thank you for the comment. By internal prior or scientific prior, we refer to the LLM’s knowledge about a scientific problem based on its general training corpus with the existing literature, i.e., what the model knows off-the-shelf about a problem without any explicit task-specific feedback or additional fine-tuning. We will make sure to further clarify this in the background section of paper (current revision is in Sec 2, L145-146).
>
> > * Why was SINDy not used as a non-LLM baseline? Why was it not mentioned in the paper at all?
> > * Please add a citation to and briefly mention how SINDy fits in to the previous work section for neural-guided deep learning methods for discovering symbolic regression:
>
> Thank you for the thoughtful question. As the focus of paper has been mostly on the recent LLM-based equation discovery approaches, our references were also centered on these approaches. We have now added reference to SINDy in the paper (L97-100) and in response to your comment, we have also conducted comparison experiments with SINDy as a new baseline. For details of experiments check App F.1. Some of the results were actually very interesting. We observed that among these four datasets, SINDy performs considerably worse than most of the state-of-the-art baselines on the stress-strain and bacterial growth datasets which contain more non-linear mathematical interactions. However, we noticed that in oscillation problems that have more linear combination of terms, SINDy obtains compatible performance with the state-of-the-art baselines and even outperforming all methods on the oscillation 2 dataset. We have provided these results and discussion in App F.2, Table 2, and updated Table 1 of the paper. We have also added citation to SINDy in the related work section (L476) in line with the reviewer's suggestion.

---

> > ### Author Response · Authors · 2025-11-27
> > **Response to Reviewer aWkA (2)**
> >
> > > * "Line 92, "and SGA": make this the start of a new sentence, ...
> > > * The first sentence of your key insight should be modified to ... " Mention that LLM frameworks don't do this explicitly, the above citation for SINDy does this exactly just without an LLM.
> > > * For "Problem Formulation" please change $x_i$ to either have a bold $\mathbf{x}_i$ or ...
> > > * Algorithm 1 should have another loop in it, since you do Stage 1 and Stage 2 multiple times to get the final solution.
> > > * Algorithm 1 is not clearly explained. What is $E$ and $e_j$? ... All of this needs to be rewritten or explained better. ...
> > > * Table 1, fix lowest score for column 1 to be LLM-SR (Mixtral) ...
> > > * Change line 439/440 "Zue et al." to an in-text citation \citet{...} ...
> > > * Line 441, there is a run-on slightly incoherent sentence ...
> > > * Line 465: change "GRPO and, evolutionary" to "GRPO and evolutionary
> > > * It's weird to have a related work section that is very similar to what you have in your introduction. Consider consolidating the sections together and moving the related work section up to the introduction.
> >
> > Thank you for the detailed and constructive comments. We have carefully addressed all points in the updated version of paper. Answering some of the questions:
> > * We have updated Algorithm 1 to clarify notation and resolve the confusion noted by reviewers, with all symbols now explicitly defined in both the algorithm and methodology sections.
> > * We have also modified the opening Related Work section and consolidated in line with reviewer suggestions.
> >
> > > * The experiment comparing with LLM-SR is not clearly stated. On one hand, it's mentioned in the main text (line 255) that 3,000 LLM calls were made. On the other hand, Figure 2 shows that there were 3,000 search candidates, implying 3,000 full loops of the method. My questions are:
> > > ** If you used 3,000 LLM calls per problem, did you count the LLM calls in GRPO (N times G=64 completions) for DecAEvolve? If so, then DecAEvolve should've had less total loops than LLM-SR due to DecAEvolve's test-time GRPO.
> > > ** If you used 3,000 loops per problem, what is the runtime comparison between LLM-SR and DecAEvolve per problem? I would expect DecAEvolve to take longer to run due to test-time model tuning using LoRA.
> > > * What is the difference in computation and runtime for your model compared to others? I would imagine it runs slower due to directional feedback (section 3.2) as well as test-time GRPO.
> >
> > That's a great question and thank you for raising this for discussion. The number of LLM calls only refers to the number of search candidates during the inference search of the discovery process. In response to reviewer's comment, we have conducted experiments in the limitted rebuttal period to better verify this. In our new exeriments, we run LLM-SR for additional 12800 samples (equivalent to the number of samples observed by GRPO in total: 64x200=12800) with Qwen2.5-7B model backbone. As it can be observed already from Figure 2 in the main paper and the results of new expeirments (Figure 6, Appendix F.2), LLM-SR is already well converged much earlier with 3000 LLM calls, and the additional sample budget equivalent to GRPO does not seem to help or significantly change its performance. This means that GRPO leverages these samples more effectively for discovery than LLM-SR.
> >
> > Also, regarding the runtime, you are correct. As we have test-time adaptation and fine-tuning of the model during the discovery process, the runtime would be naturally slower than inference-only approaches like LLM-SR or +Decomp in Figure 2. Regarding the reviewer request for runtime comparison, we would like to note that runtime is highly dependent on the underlying infrastructure, and in our current experiments, training and inference were executed on different GPU resources (remote vs. local), making a fair, apples-to-apples comparison difficult within the short rebuttal period. In response to the reviewer's comment, we have started to re-run inference-based experiments on the same remote servers used for training and will make sure to include a fair runtime comparison in the camera-ready version.
> >
> > ---
> > We hope that our rebuttal address the reviewer's concerns, and if so, they would consider updating their score. We’d be more than happy to engage in further discussions.

---

### Official Review · Reviewer_CMNd · 2025-11-01

**Soundness:** 3
**Presentation:** 4
**Contribution:** 3
**Rating:** 8
**Confidence:** 2

**Summary:**

The authors present DecAEvolve, a framework that improves current uses of LLMs for scientific equation discovery. This new framework takes advantage of given data in order to fine-tune the LLM based on data-driven rewards. This allows the LLM to reason about the contribution of each symbol, leading to an interpretable guidance towards the correct expression.

**Strengths:**

This paper is an improvement on LLM-SR, integrating reasoning about individual symbols and an evolutionary algorithm in addition to the LLM’s prior knowledge.  The paper is clearly written, with figures describing the flow of information through the evolve, adapt, and decompose sections of the framework. The evaluation is substantive, with the ablation studies showing the contribution of both the adaptation and decomposition modules.

**Weaknesses:**

I would be curious to see more about how noise in the data would affect the system’s effectiveness. Scientific data naturally will have noise due to how it’s sampled, and robustness to noise would allow this system to be more effective.

**Questions:**

1.	How does this system deal with highly correlated variables? In some problems with a high number of features, correlation between those features could cause multiple valid equations or confusion in your system in understanding the underlying symbolic structure.

2.	Can you show the performance of your model in noisy settings compared to the other baseline models?

---

> ### Author Response · Authors · 2025-11-27
> **Response to Reviewer CMNd**
>
> Thank you for dedicating your time and expertise to review our submission. Please find our responses below.
>
> > * I would be curious to see more about how noise in the data would affect the system’s effectiveness. Scientific data naturally will have noise due to how it’s sampled, and robustness to noise would allow this system to be more effective.
> > * Can you show the performance of your model in noisy settings compared to the other baseline models?
>
> Thank you for the thoughtful question. We would like to clarify our reported results already includes experiments on the Stress-Strain dataset from LLM-SR paper [1,2]. This benchmark problem leverages a real-world experimental dataset, comprising tensile tests on Aluminum 6061-T651 across a range of temperatures. The main motivation for inclusion of this benchmark in scientific equation discovery has been to test the performance of models on real noisy experimental data.
>
> > How does this system deal with highly correlated variables? In some problems with a high number of features, correlation between those features could cause multiple valid equations or confusion in your system in understanding the underlying symbolic structure.
>
> Thanks for the thoughtful question. Indeed, feature correlations are a fundamental challenge for all equation-discovery methods, not just our framework. In DecAEvolve, we try to address this to some extent by analyzing pairwise interaction contributions during the decomposition step, which helps disentangle the roles of correlated variables in the learned structure. That said, fully solving this under high collinearity still remains an open research problem in equation discovery and could be an interesting direction for future research. We have added a discussion on this point in Section 5 of the updated paper.
>
> [1] LLM-SR: Scientific Equation Discovery via Programming with Large Language Models, ICLR 2025
>
> [2] Stress-strain data for aluminum 6061-t651 from 9 lots at 6 temperatures under uniaxial and plane strain tension, in Data in Brief 2019
>
> ---
> We hope that our rebuttal address the reviewer's concerns. We’d be more than happy to engage in further discussions.

---

### Official Review · Reviewer_mkt8 · 2025-11-03

**Soundness:** 3
**Presentation:** 2
**Contribution:** 2
**Rating:** 6
**Confidence:** 4

**Summary:**

This paper builds off previous work in LLM-driven symbolic regression (LLM-SR) to address the problem of scientific equation discovery from data. It makes two modification to the LLM-SR framework:
A test-time-training component using GRPO to updated the base model weights prior to LLM evolutionary search.
2. Term-level feedback about candidate expressions during evolutionary search.
Evaluating on the same 4 synthetic examples as LLM-SR did, it shows that these changes (called DecAEvolve), outperform LLM-SR and other symbolic regression methods nearly universally across LLMs and tasks. Ablation on the 2 modifications reveal that both are beneficial, however the GRPO-based test-time-training provides most of the lift. An investigation of reward improvement with more GRPO iterations suggests that smaller, cheaper models can “catch-up” to larger models in performance given enough GRPO training.

**Strengths:**

The paper makes both a systems and technical contribution (adding GRPO-based test-time-training, and the term decomposition based feedback to evolutionary search), and ablates on both to show where the improvement comes from (primarily GRPO).
The quantitative results appear to be a significant improvement over LLM-SR (while questions of comparable computational resources remain, see below, the curves shown in Figure 2 for LLM-SR do not appear like they would ever overtake DecAEvolve.

**Weaknesses:**

This work adds two methods on top of LLM-SR; GRPO-based test-time-training (adaptation), and equation-decomposition feedback. Of these two, the decomposition is the real novel contribution, but looking at Figure 2 we can see that in most cases, the majority of the improvement over LLM-SR comes from the addition of GRPO, not the more novel contribution. Also, as near as I can tell, these charts are not taking into account the extra program samples involved in the GRPO-training, so these may not be fairly judging compute vs performance.
LLM-SR is only given one short paragraph of explanation, despite being largely reproduced as part of this method. While the mechanisms of adaptation and decomposition are (mostly) well described between the primary text and the appendix, *how* these fit into the LLM-SR framework are not. The relationship between Algorithm 1 and Figure 1 (and order of feedback through Figure 1) needs to be explicitly shown, and there are no details about the multi-island evolutionary search that is shown in Figure 1 and mentioned in the introduction.
The only results shown are error-based scores, which give no context for how large or small the effects are on the prediction results. The LLM-SR paper includes results that show the predictions of the equation discovery relative to ground truth – including such figures would go a long way to demonstrating the impact of this work.

**Questions:**

During the decomposition optimization for each term / pairwise interaction experiment, are the original parameters included in the optimization, or just the term weights?
In Fig. 2, efficiency is measured based on the # of search candidates, but on line 255 the computation budget is specified in terms of the # of LLM calls. Do the LLM calls in the budget take into account calls needed for GRPO sampling, and if not, how much more compute is needed with the GRPO-based adaptation

This work builds directly on the algorithm of LLM-SR, but does not explain that method in enough detail to be reproduced. I would suggest that the space used for the detailed explanation of GRPO in the preliminaries would be better used for a more full explanation of LLM-SR and how the modifications of DecAEvolve fit within that framework.

How important (qualitatively) are the error improvements overn LLM-SR? It would be great to include some prediction figures comparing the output of the best performing functions discovered by each framework to the ground-truth.
On line 115 it is claimed that Decomposition “enables LLMs to understand not just which hypotheses succeed, but *why* specific mathematical building blocks are effective.” Please explain this claim. It seems to me that decomposition is showing *which* mathematical building blocks are effective, but not why they are (e.g. “why” a sin block in an oscillator’s forcing function is effective would be “because the forcing function is periodic”, not that “removing the sin block increases the error by XXX”).

**Details Of Ethics Concerns:**

None.

---

> ### Author Response · Authors · 2025-11-27
> **Response to Reviewer mkt8 (1)**
>
> Thank you for dedicating your time and expertise to review our submission. Please find our responses below.
>
> > During the decomposition optimization for each term / pairwise interaction experiment, are the original parameters included in the optimization, or just the term weights?
>
> Thank you for the thoughtful question. We would like to clarify that during the decomposition, we re-optimize the remaining parameters with respect to data for each ablated structure and then compute the contribution score accordingly. So, we do not use the original parameters from the initial function structure. This is mainly to make sure that contribution score estimation is due to the structural ablations rather than suboptimal tuning or parameterization. We also have conducted experiments in response to reviewer hhwi's comment with the ablation of using original parameter values instead of their re-optimized values for ablation structures and we observe that decomposition with parameter re-optimization is empirically providing better performance during the discovery process (results are provided in Figure 7, Appendix F.3).
>
> > Of these two, the decomposition is the real novel contribution, but looking at Figure 2 we can see that in most cases, the majority of the improvement over LLM-SR comes from the addition of GRPO, not the more novel contribution.
>
> We respectfully disagree with the reviewer. Our contribution goes beyond just decomposition. The core novelty of DecAEvolve is a principled integration of decomposition, online adaptation, and evolutionary search that directly targets fundamental limitations in current LLM-based equation discovery: (1) inability to adapt to new scientific systems beyond the model’s internal priors, and (2) lack of structural guidance toward promising regions of the hypothesis space. It's also important to note that the use of GRPO here is different than general model fine-tuning with GRPO common in reasoning literature. Here, we employ GRPO for per-system, test-time adaptation that steers the model toward hypotheses better aligned with the observed numeric data of each specific scientific system.  This approach transforms the LLM from a static candidate proposer into an adaptive learner that incorporates system-specific patterns and structures during the discovery process. To the best of our knowledge, this form of integration has not been explored in prior discovery frameworks, and our experiments demonstrate consistent empirical gains over baselines.
>
> > * In Fig. 2, efficiency is measured based on the # of search candidates, but on line 255 the computation budget is specified in terms of the # of LLM calls. Do the LLM calls in the budget take into account calls needed for GRPO sampling, and if not, how much more compute is needed with the GRPO-based adaptation
> > * as near as I can tell, these charts are not taking into account the extra program samples involved in the GRPO-training, so these may not be fairly judging compute vs performance.
>
> That's a great question and thank you for raising this for discussion. Currently, the number of LLM calls only refers to the number of search candidates during the inference search of the discovery process. In response to reviewers' comments, we have conducted experiments in the rebuttal period to better verify this. In our new experiments, we run LLM-SR for additional 12800 samples (equivalent to the number of samples observed by GRPO in total: 64x200=12800) with Qwen2.5-7B model backbone. As it can be observed already from Figure 2 in the main paper and the results of new experiments (Fig 6, App F.2), LLM-SR is already well converged much earlier with 3000 LLM calls, and the additional sample budget equivalent to GRPO does not seem to help or significantly change its performance. This means that DecAEvolve leverages these samples more effectively for discovery than baselines.
>
> > On line 115 it is claimed that Decomposition “enables LLMs to understand not just which hypotheses succeed, but why specific mathematical building blocks are effective.” Please explain this claim. It seems to me that decomposition is showing which mathematical building blocks are effective, but not why they are (e.g. “why” a sin block in an oscillator’s forcing function is effective would be “because the forcing function is periodic”, not that “removing the sin block increases the error by XXX”).
>
> Thank you for the thoughtful question. You're correct that decomposition directly shows which building blocks are effective. Our claim about "why" refers to the model's implicit reasoning process: by observing the fine-grained contributions of individual sub-terms, the LLM leverages its reasoning capabilities to infer why certain components work in this context, and uses this understanding to propose better candidates in subsequent iterations.

---

> > ### Author Response · Authors · 2025-11-27
> > **Response to Reviewer mkt8 (2)**
> >
> > > How important (qualitatively) are the error improvements overn LLM-SR? It would be great to include some prediction figures comparing the output of the best performing functions discovered by each framework to the ground-truth.
> >
> > Thank you for the helpful suggestion. We agree that qualitative comparisons provide valuable context and we will make sure to include these qualitative examples in the camera-ready version of the paper.
> >
> >
> > > * This work builds directly on the algorithm of LLM-SR, but does not explain that method in enough detail to be reproduced. I would suggest that the space used for the detailed explanation of GRPO in the preliminaries would be better used for a more full explanation of LLM-SR and how the modifications of DecAEvolve fit within that framework.
> > > * LLM-SR is only given one short paragraph of explanation, despite being largely reproduced as part of this method. While the mechanisms of adaptation and decomposition are (mostly) well described between the primary text and the appendix, how these fit into the LLM-SR framework are not.
> > > * The relationship between Algorithm 1 and Figure 1 (and order of feedback through Figure 1) needs to be explicitly shown, and there are no details about the multi-island evolutionary search that is shown in Figure 1 and mentioned in the introduction.
> >
> > Thank you for the constructive comment. We've made the following revisions in response to your comment:
> > * Added Appendix A with a complete explanation of LLM-SR's methodology and clarified how DecAEvolve builds on this framework in Section 2 (p2).
> > * Revised Algorithm 1 to clearly show its relation with Figure 1, including the ordering of feedback. The updated algorithm addresses similar concerns raised by other reviewers.
> >
> > ---
> > We hope that our rebuttal address the reviewer's concerns, and if so, they would consider updating their score. We’d be more than happy to engage in further discussions.

---

### Official Review · Reviewer_p2HL · 2025-11-11

**Soundness:** 3
**Presentation:** 3
**Contribution:** 3
**Rating:** 4
**Confidence:** 4

**Summary:**

This paper focuses on automated discovery of symbolic equation with observed data of that equation. It points out two weaknesses of previous methods, adaptation and decomposition. Here adapation is about LLMs do not update their parameters for task-specific update even if after long time doing inference on a task. Decomposition is about to identify the valuable part of the equation so to more efficiently discover the full picture of the equation consistenting of several valuable parts.

**Strengths:**

1. This paper proposes to use GRPO to update the parameters of an LLM working on that research direction of scientific discovery. It has not been done in the field of automated scientific discovery before as far as I know. It would be beneficial to the community on sharing the insights of how to adapt GRPO on automated discovery of symbolic equation task, and the difference of using GRPO in this task compared with using it on other more common language tasks.

2. The idea of using "decomposition" is reasonable.

3. The experiments are sufficient.

**Weaknesses:**

1. One of the main idea, the "decomposition" has been proposed by [1] on automated discovery of chemistry scientific hypothesis with experimental feedback. The fundamental ideas are very similar, although the two papers work on scientific discovery of different tasks (math equation and chemistry). I think this submission should discuss [1] and clarify how their use of decomposition differs or extends prior formulations in this new setting.

2. It is a bit unclear on how the "adapatation" (GRPO) and "decomposition" parts "collaborate" and have the synergy effect. Currently from Algorithm 1, my understanding is that they are basically on two stages, and don't have interactions, is this understanding correct?

3. Algorithm 1 mentions many symbols such as P and E, which seems are neighther discussed in Algorithm 1 and the Method section.

4. More insights on using GRPO on the symbolic equation discovery task can be beneficial (e.g., difference with using it on other tasks)


[1] MOOSE-Chem3: Toward Experiment-Guided Hypothesis Ranking via Simulated Experimental Feedback

**Questions:**

1. Do "adapatation" (GRPO) and "decomposition" have any "interaction" (synergy) with each other?

---

> ### Author Response · Authors · 2025-11-27
> **Response to Reviewer p2HL**
>
> Thank you for dedicating your time and expertise to review our submission. Please find our responses below.
>
> > One of the main idea, the "decomposition" has been proposed by [1] on automated discovery of chemistry scientific hypothesis with experimental feedback. The fundamental ideas are very similar, although the two papers work on scientific discovery of different tasks (math equation and chemistry). I think this submission should discuss [1] and clarify how their use of decomposition differs or extends prior formulations in this new setting.
>
> Thank you for bringing this to our attention. We agree that, at a high level, both MOOSE-Chem3 [1] and our DecAEvolve framework employ a notion of “decomposition” to obtain more fine-grained feedback from hypothesis components. However, the underlying implementation and role of decomposition differ substantially. MOOSE-Chem3 operates in experimental chemistry, where hypotheses are natural-language descriptions of experimental mechanisms. Their decomposition step extracts functional chemical components (e.g., mechanistic roles) to compute a qualitative similarity measure with respect to ground-truth for experiment-guided ranking. In contrast, DecAEvolve targets different task of math equation discovery, where hypotheses are explicit symbolic expressions. Our decomposition is quantitative and analytic: we break equations into symbolic terms and operators and use an influence function to estimate each term’s contribution to predictive error and structural stability. These influence scores directly guide mutation selection, structured feedback, and the broader search dynamics. Also, decomposition is only one component in our three-module DecAEvolve framework (Decomposition, Adaptation, Evolution), whose joint interaction is central to the observed gains. To the best of our knowledge, this hybrid formulation has been explored in prior work, including [1].
>
> In response to the reviewer's comment, we have included clarification on these points and additional discussion on the comparison with [1] in the revised version of manuscript (Check Sec 3.2 and App D).
>
> > * It is a bit unclear on how the "adapatation" (GRPO) and "decomposition" parts "collaborate" and have the synergy effect. Currently from Algorithm 1, my understanding is that they are basically on two stages, and don't have interactions, is this understanding correct?
> > * Do "adapatation" (GRPO) and "decomposition" have any "interaction" (synergy) with each other?
>
> Thanks for the thoughtful question. We understand that the current version of Algorithm 1 might be confusing. To clarify, the adaptation (GRPO) and decomposition components in DecAEvolve have synergy and they are not two independent stages. During the GRPO phase, the model is first adapted/trained with prompts that include the decomposition signals. Then, during inference, the decomposition process is incorporated with evolutionary search on the adapted model.
> In response to the reviewer's comment, we have revised Algorithm 1 and the corresponding Section 3 writing to further clarify this in the updated manuscript.
>
> > Algorithm 1 mentions many symbols such as P and E, which seems are neighther discussed in Algorithm 1 and the Method section.
>
> Thank you for the comment. To address this, we have added more clarifications for Algorithm 1 in Sec3.
>
> > More insights on using GRPO on the symbolic equation discovery task can be beneficial (e.g., difference with using it on other tasks)
>
> That's a great point and thank you for raising this for discussion.  We would like to clarify that our use of GRPO differs fundamentally from its common use in general reasoning literature. Here, GRPO is not used for global model fine-tuning, but for per-system, test-time adaptation that steers the model parameters toward hypotheses better aligned with the observed scientific system. The reward is computed from how well an equation candidate explains the observed data, not from correctness or textual feedback as in typical GRPO setups. Also, the inputs/prompts used in test-time adaptation GRPO here correspond to a fixed scientific system but include dynamic in-context examples (with decomposition signal) that are sampled online from the buffer over GRPO iterations.
> In response to the reviewer's suggestion, we have also added more discussion regarding these points in the updated manuscript (Check Sec 3.1 L196-204).
>
> ---
> We hope that our rebuttal address the reviewer's concerns, and if so, they would consider updating their score. We’d be more than happy to engage in further discussions.

---

### Author Response · Authors · 2025-12-01
**Message to Area Chair and Reviewers from Authors**

Dear Area Chairs and Reviewers,

Thank you for dedicating your time and expertise to review our submission. We are grateful for the thorough and constructive comments received from reviewers for our submission. During the rebuttal period, we conducted additional experiments in response to reviewers' comments (some of which required substantial computation and unfortunately completed just before when discussion period closed for the new conference policy). We believe that we have now carefully addressed all the concerns/questions raised by reviewers in the detailed responses to each reviewer's comments and the updated manuscript. Unfortunately, we did not have the opportunity to engage with the reviewers during the discussion to address any remaining concerns.


We sincerely appreciate the time chairs and all reviewers have invested in the assessment of our submission and respectfully request the ACs and reviewers to consider our rebuttal in the final decision.

Respectfully,

Authors of Submission #22651

---

### Meta-Review · Area_Chair_3RgT · 2026-01-06

**Summary:**

This work attempts to enhance LLM-based scientific equation discovery by integrating term decomposition, GRPO-based test-time adaptation, and evolutionary search. While the core idea of addressing static LLM limitations is meaningful, the submission fails to adequately resolve key concerns raised by reviewers, leading to the rejection decision.
First, the novelty of the framework is insufficiently justified. Reviewers noted substantial overlap with prior work (e.g., LLM-SR’s architectural foundation, decomposition ideas from MOOSE-Chem3), and the claimed synergy between adaptation and decomposition remained unclear until revisions. Though the authors clarified GRPO’s task-specific application, this adjustment alone does not constitute a transformative contribution. Second, experimental design has critical flaws: the reliance on the outdated LLM-SR dataset raises potential data leakage concerns, and the lack of fair computational cost/runtime comparisons (e.g., accounting for GRPO’s extra LLM calls) undermines result credibility. While supplementary experiments (e.g., SINDy baseline comparison) were added, the failure to adopt the more mature LLM-SRBench benchmark limits generalizability. Third, method details and robustness remain unaddressed: decomposition’s susceptibility to term coupling and overfitting via BFGS refitting lacks sufficient ablation validation, and key questions (e.g., handling noisy data, highly correlated variables) were only partially answered without strong empirical support.
The authors’ rebuttal addressed some technical clarifications (e.g., algorithm notation, code availability) but did not resolve the fundamental limitations in novelty, experimental rigor, and method robustness. Thus, the submission does not meet ICLR’s standards for acceptance. We encourage the authors to refine the framework’s novel contributions, adopt standardized benchmarks, and strengthen empirical validation of key modules for future submissions.

**Reviewer Concerns:**

Reviewer mkt8:
Fairness of computational cost comparison (GRPO’s LLM calls not fully accounted for in budget; runtime comparison remains incomplete).
Lack of qualitative results (e.g., prediction vs. ground-truth figures, promised for camera-ready but not provided).
Insufficient evidence for decomposition explaining "why" building blocks work (only claims implicit model reasoning without empirical support).
Reviewer CMNd: No dedicated experiments comparing performance with baselines under controlled noisy settings (only references existing noisy dataset).
Reviewer hhwi:
Decomposition’s conceptual flaws (BFGS-induced overfitting/instability, term coupling, high computational cost) not resolved (partial ablation insufficient).
Framework novelty unconvincing (reviewer’s concern about limited distinction from LLM-SR not addressed to satisfaction).
Reliance on outdated LLM-SR dataset (data leakage concerns not fully mitigated; LLM-SRBench evaluation remains ongoing, not completed).

**Reviewer Scores:**

NA

---

### Decision · Program_Chairs · 2026-01-26

Reject